# Counterfactual Bootstrap for Robust Meta-Reinforcement Learning

**Ai Bo** [1]  **Junzhe Zhang** [1]  **M. Cenk Gursoy** [1]

## Abstract

Meta-Reinforcement Learning (Meta-RL) focuses on training policies using data collected from a variety of diverse environments. This approach enables the policy to adapt to new settings with only a few training steps. While many Meta-RL methods have demonstrated success, they often rely on the assumption that unobserved confounders can be excluded *a priori*. This paper investigates robust Meta-RL in sequential decision-making, given confounded observational data collected across multiple heterogeneous environments. We introduce a novel augmentation procedure for standard Meta-RL algorithms (e.g., MAML), which employs partial identification methods to generate posterior counterfactual trajectories from candidate environments that align with the confounded observations. These counterfactual trajectories are then used to find a policy initialization that produces strong generalization performance in the target domain. Theoretical analysis reveals that our causal Meta-RL approach is guaranteed to yield a solution that minimizes generalization loss in future inference tasks.

.

## 1. Introduction

The ability to rapidly learn and generalize across heterogeneous domains is widely regarded as a hallmark of human intelligence. Meta-learning is a critical approach to exploring how to endow AI with the capacity for fast adaptation across different environments and learning tasks (Vilalta & Drissi, 2002). Among various paradigms of meta-learning, meta reinforcement learning (meta-RL) has emerged as a crucial and popular direction, as data efficiency is essen-

tial for achieving optimal decision-making policies in RL applications. Meta-RL improves the data efficiency of RL-powered decision support systems by leveraging past data collected across different source domains to enable fast adaptation to new environments.

A variety of algorithms have been proposed for meta-RL, typically categorized by the form of inner-loop meta-parameterization: parameterized policy gradients (Finn et al., 2017; Raghu et al., 2019; Yoon et al., 2018), black box (Duan et al., 2017; Wang et al., 2016; Mishra et al., 2018), and task inference (Rakelly et al., 2019; Zintgraf et al., 2020; Humplik et al., 2019), to name a few. While these methods have achieved successes in practice, they rely on the crucial assumption that the actions observed in the data—along with the subsequent states and rewards they produce—are not simultaneously influenced by unobserved confounders. If this assumption is violated, the policies' expected returns become non-identifiable, meaning the effects cannot be determined from the available data. The following example illustrates such challenges.

**Example 1** (Challenges of Unmeasured Confounding)**.** Consider Windy Gridworlds described in Fig. 1a where the goal of the agent is to go through one of the three corridors and pick up the target key without touching the lava. For all tasks, their maps are similar except for the position and colors of the keys; each task is associated with a specific target key. At each time step $t$, the agent can take five possible actions $X_t$: `up`, `down`, `left`, `right`, or `stay-put`; there is also a wind $U_t$ blowing at each grid, following one of five directions: `east`, `south`, `west`, `north`, or `no-wind`. If the agent decides to move, its next state is shifted by both its action and the wind direction through the mechanism $S_{t+1} \leftarrow S_t + X_t + U_t$. Otherwise, the agent will stay put ($X_t \leftarrow$ `stay-put`) at its current position, regardless of the wind direction, i.e., $S_{t+1} \leftarrow S_t$. In general, the wind tempts to push the agent toward the lava; the closer the agent gets to the lava, the stronger the wind becomes.

The learning agent does not have access to the detailed system dynamics of each environment. Instead, it can observe an optimal behavioral agent that can sense the wind direction, operating in the training tasks described in Fig. 1a (left). After training, the learner will then be evaluated in the testing tasks described in Fig. 1a (left). In this meta-RL problem, the wind direction $U_t$ becomes an unobserved con-

---

[1]Department of Electrical Engineering and Computer Science, Syracuse University, Syracuse, NY, USA. Correspondence to: Ai Bo <abo100@syr.edu>.

*Proceedings of the 43rd International Conference on Machine Learning*, Seoul, South Korea. PMLR 306, 2026. Copyright 2026 by the author(s).

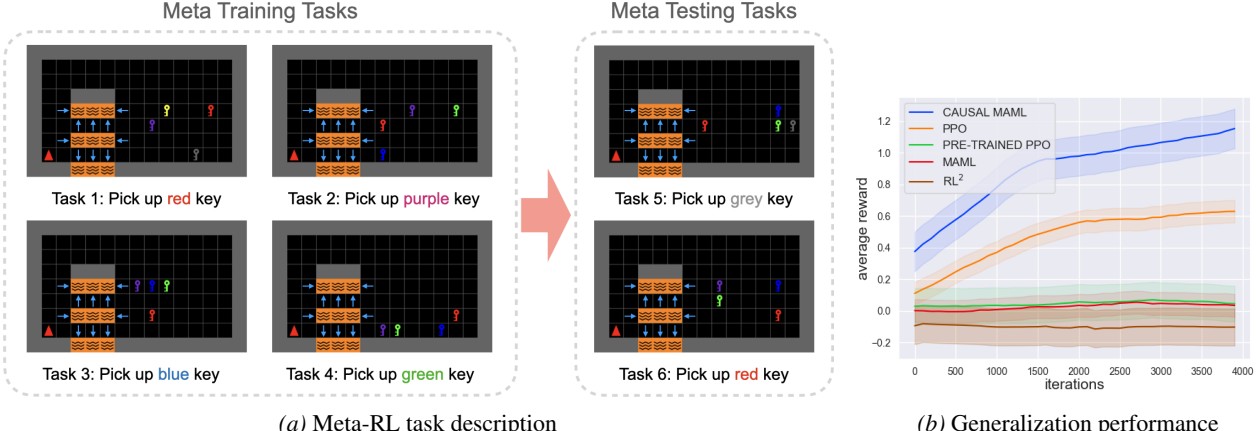

*(a)* Meta-RL task description

*(b)* Generalization performance

*Figure 1.* (a) Meta-RL tasks in a Windy Gridworld environment. Training and testing tasks are constructed by randomly generating key colors, key locations, and the target key. (b) few-shot adaptation performance comparing vanilla RL from scratch (PPO), pretrained RL (PRETRAINED-PPO), standard meta-learner (MAML), RL$^2$, and causally-enhanced meta-learner (CAUSAL-MAML).

founder affecting the observed action and state. We apply several meta-learning algorithms to this problem, including MAML (Finn et al., 2017), PPO (Schulman et al., 2017b), and RL$^2$(Duan et al., 2017) pretrained on observational data. For comparison, we also include a vanilla PPO without pretraining. Simulation results, shown in Fig. 1b, indicate that none of MAML, pretrained PPO, or RL$^2$ can outperform the vanilla PPO. We observe a significant gap between meta-learners and vanilla learners; the confounding bias in the observational data hurts meta-learners' performance. ∎

Recently, a growing body of literature has explored the nuanced interactions between causal inference theory and reinforcement learning to address data biases in the optimal decision-making under uncertainty, known as *Causal Reinforcement Learning (CRL)* (Bareinboim et al., 2024). Several algorithms have been proposed for various policy learning settings, including online learning (Bareinboim et al., 2015; Zhang & Bareinboim, 2017), off-policy learning (Kallus & Zhou, 2018; Namkoong et al., 2020; Etesami & Geiger, 2020; Zhang & Bareinboim, 2025; Li et al., 2026), imitation learning (de Haan et al., 2019; Ruan et al., 2023; 2024), and curriculum learning (Li et al., 2025b), to name a few. Few works (Dasgupta et al., 2019b;a) have explored causal structure discovery and causal reasoning using meta-learning approaches. Despite these progresses, a systematic approach for applying meta-learning to sequential decision-making tasks in finite action and state spaces with the presence of unmeasured confounding is still missing. It is unclear how one can obtain a model initialization with reasonable generalization performance when the training data is contaminated with confounding bias and potential shifts occur in the system dynamics of the testing environment.

This paper aims to address a significant gap in the field by investigating robust meta-reinforcement learning (meta-RL) using confounded observational data gathered from various

unknown Markov decision processes with similar yet distinct system dynamics. A key aspect of our approach is to employ partial causal identification, as discussed by (Balke & Pearl, 1994), alongside the representation of causal generative models introduced by (Zhang et al., 2022). This enables us to construct a feasible region of candidate models that are consistent with the observed data in each training domain. By applying gradient-based meta-learning within these feasible regions, we develop a meta-policy that performs well and generalizes well to the target environment. More specifically, our contributions are summarized as follows. (1) We introduce a novel robust augmentation procedure that leverages confounded observational data to predict non-identifiable system dynamics of the source domains while generating new counterfactual trajectories for training a meta-policy with enhanced adaptability across confounded environments. We instantiate our augmentation framework within MAML in the main paper; (2) We provide theoretical guarantees regarding the convergence of our method and detail the sample complexity necessary to obtain a good first-order stationary point approximation for the meta-RL policy. Finally, we validate our algorithm and extensions to other meta-RL methods through comprehensive simulations in RL environments. Due to space constraints, all proofs and additional experiments are provided in the Appendix.

**Notations.** We use capital letters to denote random variables ($X$), small letters for their values ($x$) and $\mathcal{X}$ for the domain of $X$. For an arbitrary set $\boldsymbol{X}$, let $|\boldsymbol{X}|$ be its cardinality. Fix indices $i, j \in \mathbb{N}$. Let $\bar{\boldsymbol{X}}_{i:j}$ stand for a sequence $\{X_i, X_{i+1}, \ldots, X_j\}$. We denote by $P(\boldsymbol{X})$ a probability distribution over variables $\boldsymbol{X}$. Similarly, $P(\boldsymbol{Y} \mid \boldsymbol{X})$ represents a set of conditional distributions $P(\boldsymbol{Y} \mid \boldsymbol{X} = \boldsymbol{x})$ for all realizations $\boldsymbol{x}$. We consistently use $P(\boldsymbol{x})$ as abbreviations of probabilities $P(\boldsymbol{X} = \boldsymbol{x})$; so does $P(\boldsymbol{Y} = \boldsymbol{y} \mid \boldsymbol{X} = \boldsymbol{x}) = P(\boldsymbol{y} \mid \boldsymbol{x})$. Finally, $\mathbb{1}_{\boldsymbol{Z} = \boldsymbol{z}}$ is an indicator function that

returns 1 if event $\boldsymbol{Z} = \boldsymbol{z}$ holds true; otherwise, it returns 0.

## 2. Meta-Reinforcement Learning with Unmeasured Confounding

We will consider the sequential decision-making setting in which the agent selects a sequence of actions to optimize subsequent rewards. Throughout this paper, we will focus on a generalized family of confounded MDPs (Zhang & Bareinboim, 2016; Kallus & Zhou, 2020; Bennett et al., 2021) where the unobserved confounders are not assumed away *a priori*, and the learner does not deliberately control the behavioral policy that generates the observational data.

**Definition 1.** A Confounded Markov Decision Process (CMDP) $\mathcal{M}$ is a tuple of $\langle \mathcal{S}, \mathcal{X}, \mathcal{Y}, \mathcal{U}, \mathcal{F}, P \rangle$ where (1) $\mathcal{S}, \mathcal{X}, \mathcal{Y}$ are, respectively, the spaces of observed states, actions, and rewards; (2) $\mathcal{U}$ is the space of unobserved exogenous noise; (3) $\mathcal{F}$ is a set consisting of the transition function $f_S : \mathcal{S} \times \mathcal{X} \times \mathcal{U} \mapsto \mathcal{S}$, behavioral policy $f_X : \mathcal{S} \times \mathcal{U} \mapsto \mathcal{X}$, and reward function $f_Y : \mathcal{S} \times \mathcal{X} \times \mathcal{U} \mapsto \mathcal{Y}$; (4) $P$ is an exogenous distribution over the domain $\mathcal{U}$.

Throughout this paper, we will consider CMDPs with a finite horizon $H < \infty$; we consistently assume the action domain $\mathcal{X}$ and the state domain $\mathcal{S}$ to be discrete and finite; the reward domain $\mathcal{Y}$ is bounded in a real interval $[a, b] \subset \mathbb{R}$. A policy $\pi$ in a CMDP $\mathcal{M}$ is a decision rule $\pi(x_t \mid s_t)$ mapping from state to a distribution over action domain $\mathcal{X}$. An intervention $\mathrm{do}(\pi)$ is an operation that replaces the behavioral policy $f_X$ in CMDP $\mathcal{M}$ with the policy $\pi$ (Pearl, 2000, Ch. 5). Let $\mathcal{M}_\pi$ be the submodel induced by intervention $\mathrm{do}(\pi)$. A realization of states and actions is called a trajectory and can be written as $\tau = (\bar{\boldsymbol{x}}_{1:H}, \bar{\boldsymbol{s}}_{1:H}, \bar{\boldsymbol{y}}_{1:H})$. The interventional distribution $P_\pi(\bar{\boldsymbol{X}}_{1:H}, \bar{\boldsymbol{S}}_{1:H}, \bar{\boldsymbol{Y}}_{1:H})$ is defined as the joint distribution over observed variables in thus post-interventional submodel $\mathcal{M}_\pi$,

$$P_\pi(\bar{\boldsymbol{x}}_{1:H}, \bar{\boldsymbol{s}}_{1:H}, \bar{\boldsymbol{y}}_{1:H}) \tag{1}$$
$$= P(s_1) \prod_{t=1}^{H} \left( \pi(x_t \mid s_t) \mathcal{T}(s_t, x_t, s_{t+1}) \mathcal{R}(s_t, x_t, y_t) \right)$$

where the transition $\mathcal{T}$ and reward distribution $\mathcal{R}$ are given by $\mathcal{T}(s_t, x_t, s_{t+1}) = \int_{\mathcal{U}} \mathbb{1}_{s_{t+1} = f_S(s_t, x_t, u_t)} P(u_t)$ and $\mathcal{R}(s_t, x_t, y_t) = \int_{\mathcal{U}} \mathbb{1}_{y_t = f_Y(s_t, x_t, u_t)} P(u_t)$. We write the reward function $\mathcal{R}(s, x) = \sum_y y \mathcal{R}(s, x, y)$.

A common objective for an RL agent is to optimize its cumulative return $J_\pi = \mathbb{E}_\pi \left[ \sum_{t=1}^{H} \gamma^{t-1} Y_t \right]$ where $0 \leq \gamma \leq 1$ is the discount factor. When detailed parametrizations of the underlying distribution and function are provided, there exist standard planning methods to compute the optimal policy (Bellman, 1966; Sutton & Barto, 1998). However, in many practical scenarios, the detailed knowledge of the environments is often not fully available. In this paper, we

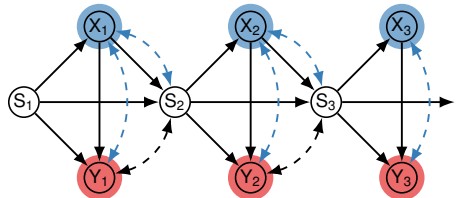

*Figure 2.* Causal diagram representing the data-generating mechanisms in a Confounded Markov Decision Process.

consider learning settings where the agent has access to the observational data in CMDPs, generated by demonstrators following behavioral policies. Specifically, for every time step $t = 1, \ldots, H$, the environment first draws an exogenous noise $U_t$ from the distribution $P(\mathcal{U})$; the demonstrator then performs an action $X_t \leftarrow f_X(S_t, U_t)$ following the behavioral policy $f_X$, receives a subsequent reward $Y_t \leftarrow r_t(S_t, X_t, U_t)$, and moves to the next state $S_{t+1} \leftarrow f_S(S_t, X_t, U_t)$. The observed trajectories are summarized as the observational distribution $P(\bar{\boldsymbol{X}}_{1:H}, \bar{\boldsymbol{S}}_{1:H}, \bar{\boldsymbol{Y}}_{1:H})$.

Fig. 2 shows the causal diagram $\mathcal{G}$ (Bareinboim et al., 2022) describing the generative process of the observational data in CMDPs, where nodes represent observed variables $X_t, S_t, Y_t$, and arrows represent the functional relationships $f_X, f_S, f_Y$ among them. Exogenous variables $U_t$ are often not explicitly shown; bi-directed arrows $X_t \leftarrow\rightarrow Y_t$ and $X_t \leftarrow\rightarrow S_{t+1}$ (highlighted in blue) indicate the presence of an unobserved confounder (UC) $U_t$ affecting the action, state, and reward simultaneously. The presence of these unobserved confounders violates the condition of no unmeasured confounding (Robbins, 1985; Bareinboim et al., 2024), posing challenges for various policy learning tasks, including meta-RL (Finn et al., 2017).

**Meta-Reinforcement Learning.** Let $\mathcal{B} = \{\mathcal{M}_i\}_{i=1}^{B}$ be the set of CMDPs representing different RL tasks. We assume these CMDPs are drawn from a distribution $\rho$ (which Nature will draw samples from). The detailed parametrizations of exogenous distribution $P_i$ and structural functions $\mathcal{F}_i$ for these CMDPs $\mathcal{M}_i$ generally differ from one another. We will consistently use $\mathcal{D}_{\mathrm{obs}}^i$ to denote trajectories collected passively observing a demonstrator operating in the model $\mathcal{M}_i$, following the observational distribution $P(\bar{\boldsymbol{X}}_{1:H}, \bar{\boldsymbol{S}}_{1:H}, \bar{\boldsymbol{Y}}_{1:H})$. Similarly, we use $\mathcal{D}_{\mathrm{exp}}^i$ to denote the experimental trajectories collected from performing interventions $\mathrm{do}(\pi_i)$ in the model $\mathcal{M}_i$ following policy $\pi_i$.

To demonstrate our general data augmentation technique, we apply it to a well-known meta-RL method, MAML (Finn et al., 2017). The goal of MAML is to learn a policy $\pi$ that peforms well as an initialization for learning a new unseen task $\mathcal{M}_i$ when the learner has a budget for running a few steps of gradient descent. To search over the space of all policies, we assume these policies are parametrized with

$\theta \in \mathbb{R}^d$. We denote the policy corresponding to parameter $\theta$ by $\pi(\cdot; \theta)$ and the expected return corresponding to this policy $\pi(\cdot; \theta)$ in a model $\mathcal{M}_i$ by $J_i(\theta)$. For simplicity, we focus on finding an initialization $\theta$ such that, after observing a new CMDP $\mathcal{M}_i$, one gradient step would lead to a good approximation for the minimizer of $J_i(\theta)$. We can formulate this learning goal as follows,

$$\max_\theta F(\theta) := \mathbb{E}_{\mathcal{M}_i \sim \rho} \left[ J_i \left( \theta + \alpha \nabla J_i(\theta) \right) \right], \quad (2)$$

where the step size $\alpha$ is a hyper-parameter that controls the magnitude of the gradient ascent update.

In practice, since the detailed system dynamics of the target CMDP $\mathcal{M}_i$ are unknown, one must estimate the policy gradient $\nabla J_i(\theta)$ from empirical samples collected from the environment. Unbiased estimation methods have been proposed (Finn et al., 2017; Fallah et al., 2020) to approximate the gradient when the learner could directly intervene in the environment. Specifically, the learner will intervene in the CMDP $\mathcal{M}_i$, collect experimental data $\mathcal{D}_{\exp}^i$, evaluate the stochastic gradient $\tilde{\nabla} J_i(\theta, \mathcal{D}_{\exp}^i)$ from the batch, and solve for the optimal solution $\theta$ of Eq. (2) by replacing the gradient $\nabla J_i(\theta)$ with $\tilde{\nabla} J_i(\theta, \mathcal{D}_{\exp}^i)$. When $\tilde{\nabla} J_i(\theta, \mathcal{D}_{\exp}^i)$ is an unbiased estimator, this meta-RL approach has demonstrated success and achieved an optimal initialization $\theta^*$.

However, challenges could arise when the agent does not have access to directly intervene in the task $\mathcal{M}_i$. Without realizing the discrepancy between the observational $\mathcal{D}_{\text{obs}}^i$ and experimental data $\mathcal{D}_{\exp}^i$, a naive learner might use $\mathcal{D}_{\text{obs}}^i$ as if it were $\mathcal{D}_{\exp}^i$, and proceed with the original MAML method. This procedure leads to the following optimization problem:

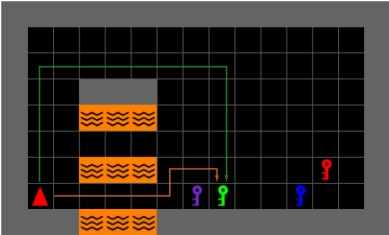

Figure 3. Comparing two routes (long and short) to reach the target green key.

$$\max_\theta \tilde{F}(\theta) = \mathbb{E}_{\mathcal{M}_i \sim \rho} \left[ \mathbb{E}_{\mathcal{D}_{\text{obs}}^i} \left[ J_i \left( \theta + \alpha \tilde{\nabla} J_i(\theta, \mathcal{D}_{\text{obs}}^i) \right) \right] \right]. \quad (3)$$

Among the above quantities, $\tilde{\nabla} J_i(\theta, \mathcal{D}_{\text{obs}}^i)$ is the stochastic gradient evaluated from the observational data $\mathcal{D}_{\text{obs}}^i$. Generally, when the unobserved confounding exists, the underlying system dynamics are underdetermined (i.e., non-identifiable) from the observational data (Kallus & Zhou, 2018; Zhang & Bareinboim, 2025). Consequently, the stochastic gradient $\tilde{\nabla} J_i(\theta, \mathcal{D}_{\text{obs}}^i)$ is no longer an unbiased estimate of $\nabla J_i(\theta)$, and solving the optimization in Eq. (3) yields a solution $\theta$ with sub-optimal behavior.

**Example 2** (Windy Gridworlds continued). Consider the meta-reinforcement learning task of windy gridworlds de-

scribed in Fig. 1a. In this scenario, the wind direction $U_t$ serves as an unobserved confounder that influences the observed action $X_t$, the subsequent reward $Y_t$, and the next state $S_{t+1}$. This introduces spurious correlations in the observational data, causing some trajectories to appear associated with higher returns. For example, Fig. 3 illustrates two observed trajectories leading to the target green key. The shorter orange route is risky, as it requires navigating a narrow passage between lava tiles. The demonstrator, able to sense the wind direction, can stop when pushed toward the lava and thus consistently take the short route to reach the key. However, the learner cannot sense the wind and cannot choose the right moment to stop. If the learner naively updates its policy using the stochastic gradient $\tilde{\nabla} J_i(\theta, \mathcal{D}_{\text{obs}}^i)$ derived from the observational data, it will not accurately recover the actual gradient $\nabla J_i(\theta)$. Instead, it will overestimate the value of risky short-route trajectories, leading to sub-optimal performance. In contrast, the learner should consider taking the longer but safer upper passage, which is more reliable even in windy conditions. ∎

To better highlight the difference between the optimal policy initialization for meta-RL in Eq. (2) and the biased solution obtained by naively applying standard MAML in Eq. (3) with confounded observations, we

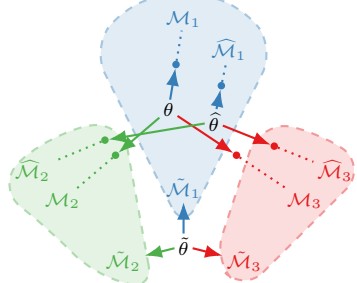

Figure 4. Comparing the optimal solution $\theta$ of Eq. (2) and solutions obtained by naive meta-RL $\tilde{\theta}$ (Eq. (3)) and the causally enhanced approach $\hat{\theta}$ (Eq. (4)).

consider an example with three equally likely CMDPs $\mathcal{M}_1, \mathcal{M}_2, \mathcal{M}_3$; see Fig. 4. For each sampled CMDP $\mathcal{M}_i$, the dashed shade represents the equivalence class of environments $\tilde{M}_i$ compatible with the same observational data. When unmeasured confounding exists, one cannot distinguish between the actual task $\mathcal{M}_i$ and the other task $\tilde{\mathcal{M}}_i$, and these models could have significantly different system dynamics. If one is not aware of this difference and naively applies MAML gradient update using confounded observations, the algorithm will converge to the alternative task $\tilde{\mathcal{M}}_i$ in the equivalence. When the confounding bias is significant and $\tilde{\mathcal{M}}_i$ deviates from the actual task $\mathcal{M}_i$, the obtained solution $\tilde{\theta}$ could deviate from the optimal $\theta$ and fail to generalize to all environments.

## 3. Confounding Robust Meta-RL

A natural question arising at this point is how to perform robust meta-RL in the face of unmeasured confounding in

the observational data. Our analysis so far suggests that when the no-unmeasured-confounding condition does not hold, it is infeasible to obtain an unbiased stochastic gradient for the policy update, thereby preventing the recovery of the optimal meta-policy in Eq. (2). For the remainder of this paper, we show that this is not the case by proposing a novel counterfactual data augmentation that enables meta-RL algorithms to be robust against confounded observations.

Note that CMDP tasks $\mathcal{M}_i$ are drawn from a prior distribution $\rho$. Our discussion begins with a meta-RL approach assuming access to an oracle capable of sampling the posterior tasks $\widehat{\mathcal{M}}_i \sim \rho(\mathcal{M} \mid \mathcal{D}_{\text{obs}}^i)$ conditioned on the observational data $\mathcal{D}_{\text{obs}}^i$. We will then relax this assumption by providing a practical Monte-Carlo approach to sample the posterior distribution. Specifically, after observing a CMDP task $\mathcal{M}_i$ and receiving the observational data $\mathcal{D}_{\text{obs}}^i$, instead of evaluating the gradient $\nabla J_i(\theta)$ from confounded observations, our causal learner will sample an alternative model $\widehat{\mathcal{M}}_i$ compatible with the same observations from the oracle $\rho(\mathcal{M} \mid \mathcal{D}_{\text{obs}}^i)$. The causal meta-learner will then interact with this posterior model $\widehat{\mathcal{M}}_i$ and collect the subsequent experimental data $\widehat{\mathcal{D}}_{\text{exp}}^i$. Finally, the causal learner performs the stochastic gradient update $\widehat{\nabla} J_i(\theta, \widehat{\mathcal{D}}_{\text{exp}}^i)$ using the posterior experimental data. This augmented meta-RL procedure could be formalized as the following optimization program:

$$
\max_\theta \widehat{F}(\theta) \tag{4}
$$
$$
= \mathbb{E}_{\mathcal{M}_i \sim \rho} \left[ \mathbb{E}_{\mathcal{D}_{\text{obs}}^i} \left[ \mathbb{E}_{\widehat{\mathcal{D}}_{\text{exp}}^i} \left[ J_i \left( \theta + \alpha \widehat{\nabla} J_i(\theta, \widehat{\mathcal{D}}_{\text{exp}}^i) \right) \right] \right] \right].
$$

In the above equation, computing the posterior experimental data $\widehat{\mathcal{D}}_{\text{exp}}^i$ conditioned on the observational trajectories $\mathcal{D}_{\text{obs}}^i$ can be seen as performing a counterfactual query. That is, *"given the observed trajectories (collected from the demonstrator), what would the outcome be had I personally taken the same route as the observed one (or exploring an alternative route)?"* Henceforth, we will consistently refer to this augmentation step as the *counterfactual bootstrap*.

Fig. 4 illustrates this intuition by comparing the solution $\widehat{\theta}$ of Eq. (4) to the optimal solution of Eq. (2). Here, $\widehat{\theta}$ is a meta-policy computed using the counterfactual CMDPs drawn from the oracle $\widehat{\mathcal{M}}_i \sim \rho(\mathcal{M} \mid \mathcal{D}_{\text{obs}}^i)$. Since the oracle provides access to the posterior over all tasks conditioned on observed trajectories, the solution $\widehat{\theta}$ is a consistent estimate of the optimal solution in expectation, thereby leading to a reasonable generalization performance.

### 3.1. Counterfactual Bootstrap.

The causal meta-reinforcement learning (meta-RL) method discussed earlier depends on having oracle access to the posterior distribution $\rho(\mathcal{M}_i \mid \mathcal{D}_{\text{obs}}^i)$, which is conditioned on the confounded observations. However, evaluating this

posterior can be difficult in practice because we lack detailed information about the prior distribution $\rho(\mathcal{M})$ over potential tasks. One possible solution is to define a non-informative prior $\widehat{\rho}$ to serve as an approximation of the actual prior $\rho$. However, constructing such a prior $\widehat{\rho}$ is complicated, as we do not know the specific parametric forms of the distribution $P$ and the structural functions $\mathcal{F}$ for the underlying CMDPs. To address this challenge, we will utilize a parametric family of canonical causal models introduced by (Zhang et al., 2022), which limits the cardinality of the latent domain to that of the observed state-action space.

**Definition 2.** A canonical CMDP $\mathcal{M}$ is a CMDP $\langle \mathcal{S}, \mathcal{X}, \mathcal{Y}, \mathcal{U}, \mathcal{F}, P \rangle$ where its the cardinality of the exogenous domain $\mathcal{U}$ is bounded by $|\mathcal{U}| \leq 2(|\mathcal{S} \times \mathcal{X}| + |\mathcal{S} \times \mathcal{X} \times \mathcal{S}| + |\mathcal{S} \times \mathcal{X} \times \mathcal{Y}|)$.

For a canonical CMDP, the latent cardinality of the exogenous domain is bounded by a linear function of the cardinality of the observed state-action space. For standard CMDPs with discrete states and actions, the latent exogenous domain is also discrete and finite. A critical property of canonical causal models is that they preserve the values of all the observational and interventional distributions defined by the original, unrestricted causal models using a finite number of latent states (Zhang et al., 2022, Thm. 2.4).

**Corollary 1.** *For an arbitrary CMDP $\mathcal{M}$, there exists a canonical CMDP $\mathcal{N}$ such that for any finite horizon $H < \infty$ and any policy $\pi$, $P(\bar{\boldsymbol{x}}_{1:H}, \bar{\boldsymbol{s}}_{1:H}, \bar{\boldsymbol{y}}_{1:H}; \mathcal{M}) = P(\bar{\boldsymbol{x}}_{1:H}, \bar{\boldsymbol{s}}_{1:H}, \bar{\boldsymbol{y}}_{1:H}; \mathcal{N})$ and $P_\pi(\bar{\boldsymbol{x}}_{1:H}, \bar{\boldsymbol{s}}_{1:H}, \bar{\boldsymbol{y}}_{1:H}; \mathcal{M}) = P_\pi(\bar{\boldsymbol{x}}_{1:H}, \bar{\boldsymbol{s}}_{1:H}, \bar{\boldsymbol{y}}_{1:H}; \mathcal{N})$.*

Corol. 1 implies that for meta-RL tasks from the observational data over discrete domains, one could assume the latent states of the underlying CMDPs to be discrete and finite without loss of generality. This latent space reduction simplifies the construction of the approximate prior $\widehat{\rho}$. The posterior sampling of counterfactual models thus follows the procedure introduced in (Zhang et al., 2022).

In practice, direct sampling from posterior CMDP models can be computationally challenging as the cardinality of the observed domain increases. In this case, one could specify the closed-form characterization of the boundary conditions for the feasible regions containing candidate CMDPs compatible with the data. Specifically, for any CMDP $\mathcal{M}_i \sim \widehat{\rho}(\mathcal{M} \mid \mathcal{D}_{\text{obs}}^i)$, its transition probabilities $\mathcal{T}_i$ can be bounded following the strategy introduced in (Manski, 1990). That is, for any state-action-state tuple $s, x, s'$,

$$
\mathcal{T}_i(s, x, s') \geq p(s' \mid x, s) p(x \mid s) \tag{5}
$$
$$
\mathcal{T}_i(s, x, s') \leq p(s' \mid x, s) p(x \mid s) + p(\neg x \mid s) \tag{6}
$$

where $p(s' \mid x, s)$ and $p(x \mid s)$ are functions computable from the observational distribution such that $p(s' \mid x, s) = P(S_{t+1} = s' \mid X_t = x, S_t = s)$, $p(x \mid s) = P(X_t = x \mid$

$S_t = s$) and $p(\neg x \mid s) = 1 - P(x \mid s)$. Similarly, the reward function $\mathcal{R}_i$ for any CMDP $\mathcal{M}_i \sim \widehat{\rho}(\mathcal{M} \mid \mathcal{D}_{\text{obs}}^i)$ can be bounded from the observational distribution and is given by, for any state-action pair $s, x$,

$$\mathcal{R}_i(s, x) \geq r(s, x)p(x \mid s) + ap(\neg x \mid s) \tag{7}$$

$$\mathcal{R}_i(s, x) \leq r(s, x)p(x \mid s) + bp(\neg x \mid s) \tag{8}$$

where $r(s, x)$ is the nominal reward function given by $r(s, x) = \mathbb{E}[Y_t \mid S_t = s, X_t = x]$, and $[a, b]$ is an interval containing the reward signal $Y_t$. To approximate counterfactual CMDPs drawn from the posterior $\widehat{\rho}(\mathcal{M} \mid \mathcal{D}_{\text{obs}}^i)$, one could sample transition probabilities $\mathcal{T}_i$ and reward function $\mathcal{R}_i$ uniformly from the bounds in Eqs. (5) and (6) and Eqs. (7) and (8) respectively. Since we have access to the closed-form solutions, these bounds can be estimated from observational data using functional approximations.

### 3.2. Causal MAML.

We will demonstrate our counterfactual data augmentation in the context of MAML (Finn et al., 2017), a popular gradient-based meta-RL method. However, we note that our counterfactual bootstrap could be applied to other meta-RL methods to combat confounding bias. For details of these methods and their evaluations, we refer readers to Sec. B.2.

Details of the augmented algorithm, called CAUSAL-MAML, are described in Alg. 1. Similar to MAML, its training contains an inner loop and an outer loop. More specifically, at Line 3, Nature (e.g., a system designer) selects a collection of source meta-training CMDP tasks $\mathcal{B} = \{\mathcal{M}_i\}$ following the distribution $\rho$. For every CMDP $\mathcal{M}_i$ in the inner training loop, the learner observes its trajectories (generated by a demonstrator) and obtains the observational data $\mathcal{D}_{\text{obs}}^i$ (Line 5). It then constructs an approximate posterior $\widehat{\rho}(\mathcal{M} \mid \mathcal{D}_{\text{obs}}^i)$ and draws an alternative environment $\widehat{\mathcal{M}}_i$ from the posterior, following the counterfactual bootstrap procedure described previously. The learner simulates interventions following the current policy estimate $\pi(\cdot \mid \cdot; \theta)$ in the sampled CMDP $\widehat{\mathcal{M}}_i$ and collects experimental trajectories $\widehat{\mathcal{D}}_{\text{exp,in}}^i$ (Line 7). It then computes the inner stochastic gradient $\widehat{\nabla}_\theta J_i(\theta, \widehat{\mathcal{D}}_{\text{exp,in}}^i)$ using the collected experimental trajectories. Formally, given finite experimental trajectories $\widehat{\mathcal{D}}_{\text{exp}}$, we define the stochastic gradient $\widehat{\nabla}_\theta J_i(\theta, \widehat{\mathcal{D}})$ as:

$$\widehat{\nabla}_\theta J_i(\theta, \widehat{\mathcal{D}}) \tag{9}$$
$$= \frac{1}{|\widehat{\mathcal{D}}|} \sum_{\tau \in \widehat{\mathcal{D}}} \sum_{t=0}^H \nabla_\theta \log \pi(x_t \mid s_t; \theta) \sum_{t'=t}^H \gamma^{t'} \mathcal{R}_i(s_{t'}, x_{t'})$$

At Lines 9-10, the learner updates the parameter $\theta_i$ of an adapted policy $\pi(\cdot \mid \cdot; \theta_i)$ and uses this policy to subsequently interact with the sampled CMDP $\widehat{\mathcal{M}}_i$ to generate outer-loop experimental trajectories $\widehat{\mathcal{D}}_{\text{exp,o}}^i$. After completing the inner training loop for every source task, the learner

finally enters the outer-loop update and adjusts the parameter $\theta$ using the gradient of meta-RL objective function $\widehat{\nabla}_\theta F(\theta)$ evaluated at the adapted parameter $\theta_i$ and the outer-loop trajectories $\widehat{\mathcal{D}}_{\text{exp,o}}^i$. Formally, the stochastic gradient of the meta-objective function is defined as follows:[1]

$$\widehat{\nabla}_\theta F(\theta) \tag{10}$$
$$= \frac{1}{|\mathcal{B}|} \sum_{i \in \mathcal{B}} \left( \left( I + \alpha \widehat{\nabla}_\theta^2 J_i(\theta, \widehat{\mathcal{D}}_{\text{exp,in}}^i) \right) \widehat{\nabla}_\theta J_i \left( \theta_i, \widehat{\mathcal{D}}_{\text{exp,o}}^i \right) \right.$$
$$\left. + \widehat{J}_i \left( \theta_i, \widehat{\mathcal{D}}_{\text{exp,o}}^i \right) \sum_{\tau \in \widehat{\mathcal{D}}_{\text{exp,in}}^i} \sum_{t=0}^H \nabla_\theta \log \pi(x_t \mid s_t; \theta) \right).$$

Among quantites in the above equation, $I$ is an identity matrix; $\widehat{J}_i(\theta_i, \widehat{\mathcal{D}}_{\text{exp,o}}^i)$ is the empirical mean estimate of the expected return for a policy $\pi(\cdot \mid \cdot; \theta_i)$ evaluated from outer-loop trajectories $\widehat{\mathcal{D}}_{\text{exp,o}}^i$. $\widehat{\nabla}_\theta^2 J_i(\theta, \widehat{\mathcal{D}})$ is policy Hessian estimate for sampled CMDP $\widehat{\mathcal{M}}_i$ defined as

$$\widehat{\nabla}_\theta^2 J_i(\theta, \widehat{\mathcal{D}}) \tag{11}$$
$$= \frac{1}{|\widehat{\mathcal{D}}|} \sum_{\tau \in \widehat{\mathcal{D}}} \left( \left( \sum_{t=0}^H \nabla_\theta \log \pi(x_t \mid s_t; \theta) \Psi_t \right) \right.$$
$$\left. \times \nabla_\theta \log p_i(\tau; \theta) + \sum_{t=0}^H \nabla_\theta^2 \log \pi(x_t \mid s_t; \theta) \Psi_t \right)$$

with the interventional probability $p_i(\tau; \theta) = P_{\pi(\cdot \mid \cdot; \theta)}(\tau)$. It can be verified that if all the gradients and Hessians in the outer-loop update were exact, then the outcome of the update would be equivalent to the outcome of the gradient ascent update for the objective $\widehat{F}(\theta)$ (Fallah et al., 2021).

### 3.3. Convergence of Causal MAML

This section will analyze the asymptotic properties of our proposed CAUSAL-MAML algorithm and provide theoretical guarantees for the computational complexity of its convergence. Our discussion begins with introducing some necessary conditions on the smoothness of the hypothesis class containing the candidate policy networks.

**Assumption 1.** The gradient and Hessian of logarithmic policy are bounded; that is, there exist constants $G, L \in \mathbb{R}$ such that, for any state $s \in \mathcal{S}$, action $x \in \mathcal{X}$, and parameter $\theta \in \mathbb{R}^d$, we have $\|\nabla_\theta \log \pi_\theta(x \mid s; \theta)\| \leq G$ and $\|\nabla_\theta^2 \log \pi(x \mid s; \theta)\| \leq L$.

**Assumption 2.** The Hessian of logarithmic policy is $K$-Lipschitz continuous; that is, there exists a real constant $K > 0$ such that for all parameters $\theta_1, \theta_2 \in \mathbb{R}^d$, state $s \in \mathcal{S}$ and action $x \in \mathcal{X}$, we have $\|\nabla_\theta^2 \log \pi(x \mid s; \theta_1) - \nabla_\theta^2 \log \pi(x \mid s; \theta_2)\| \leq K\|\theta_1 - \theta_2\|$.

---

[1]For simplicity, we assume that all experimental trajectories $\widehat{\mathcal{D}}_{\text{exp,in}}^i$ and $\widehat{\mathcal{D}}_{\text{exp,o}}^i$ have the same size $D$.

**Algorithm 1:** CAUSAL-MAML

**1 Require:** Initial parameter $\theta$, an approximate prior over CMDPs $\widehat{\rho}(\mathcal{M})$

**2 while** not done **do**

**3**     Nature samples a batch of CMDP tasks $\mathcal{B} = \{\mathcal{M}_i\}_{i=1}^{B}$ from distribution $\rho(\mathcal{M})$

**4**     **for** all task $\mathcal{M}_i \in \mathcal{B}$ **do**

**5**        Sample observation trajectories $\mathcal{D}_{\text{obs}}^i$ in environment $\mathcal{M}_i$

**6**        Sample a new environment $\widehat{\mathcal{M}}_i$ from the posterior $\widehat{\rho}(\mathcal{M} \mid \mathcal{D}_{\text{obs}}^i)$

**7**        Sample experimental trajectories $\widehat{\mathcal{D}}_{\text{exp,in}}^i$ using agent policy $\pi(\cdot \mid \cdot; \theta)$ in environment $\widehat{\mathcal{M}}_i$

**8**        Compute inner gradient $\widehat{\nabla}_\theta J_i(\theta, \widehat{\mathcal{D}}_{\text{exp,in}}^i)$ using dataset $\widehat{\mathcal{D}}_{\text{exp,in}}^i$ following Eq. (9)

**9**        Set adapted parameter $\theta_i = \theta + \alpha\widehat{\nabla}_\theta J_i(\theta, \widehat{\mathcal{D}}_{\text{exp,in}}^i)$

**10**       Sample experimental dataset $\mathcal{D}_{\text{exp,o}}^i$ using adapted policy $\pi(\cdot \mid \cdot; \theta_i)$ in environment $\widehat{\mathcal{M}}_i$

**11**     **end**

**12**

**13**     Update parameter $\theta \leftarrow \theta + \beta\widehat{\nabla}_\theta F(\theta)$ following Eq. (10)

**14 end**

Assumption 1 states that the gradient and Hessian of the logarithmic policy distribution are bounded, and Assumption 2 implies that the Hessian of the logarithmic policy distribution is Lipschitz continuous. In practice, these assumptions generally hold for some common policy classes, including neural networks with softmax layers (Bridle, 1990) and smooth activation functions (Dugas et al., 2000).

In practice, the meta-RL problem of Fig. 4 is generally non-convex. Due to this reason, we will focus on finding a policy initialization that satisfies the first-order optimality condition. Formally, a solution $\theta_\epsilon \in \mathbb{R}^d$ is called an $\epsilon$-approximate first-order stationary point ($\epsilon$-FOSP), if it satisfies $\|\nabla F(\theta_\epsilon)\| \leq \epsilon$, i.e., it approximates a local optimum of the meta-objective function. Our following result establishes the convergence of the proposed causal meta-learner.

**Theorem 1.** *Consider the case that $\alpha \in (0, 1/\eta_H]$ and $\beta \in (0, 1/L_H]$. For any $\epsilon \in (0, 1)$, CAUSAL-MAML finds a solution $\theta_\epsilon$ satisfying $E[\|\nabla_\theta F(\theta_\epsilon)\|^2] \leq 2L_G^2 L_H \beta B^{-1} D^{-1} + \epsilon^2$, after running at most for $\mathcal{O}(1)(b-a)(1-\gamma)^{-1}\beta^{-1}\min(\epsilon^{-2}, BDL_G^{-2}L_H^{-1}\beta^{-1}/2)$ iterations.*

Thm. 1 implies that the causal meta-learner is guaranteed to find a local-optimum solution for the policy initialization of

Fig. 4 with a sufficient number of iterations and trajectories. It also allows us to characterize the computational complexity of CAUSAL-MAML for finding an $\epsilon$-FOSP solution. Fix an error rate $\epsilon > 0$. The convergence condition of Thm. 1 implies two possible settings: (1) when $\beta = 1/L_G$, our CAUSAL-MAML requires at least $\mathcal{O}(\epsilon^{-2})$ iterations, with a total number of $\epsilon^{-2}$ trajectories per iteration to reach an $\epsilon$-FOSP solution; and (2) $\beta = \epsilon^{-2}$, CAUSAL-MAML requires at least a total number of $\mathcal{O}(\epsilon^{-4})$ iterations, with $\mathcal{O}(1)$ trajectories per iteration. In both cases, the total number of stochastic gradient evaluations is $\mathcal{O}(\epsilon^{-4})$.

## 4. Experiments

In this section, we validate the proposed counterfactual augmentation in various meta-RL tasks in the Windy Grid-worlds (Li et al., 2025a; Zhang & Bareinboim, 2025), which is adapted from the Minigrid environment (Chevalier-Boisvert et al., 2023). In these environments,

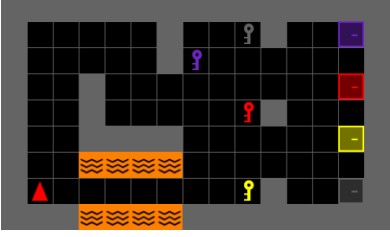

*(a)* Go-To-Door

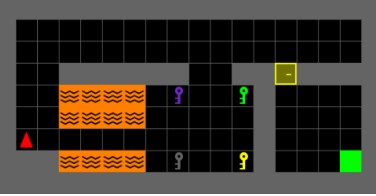

*(b)* Go-To-Goal

*Figure 5.* Meta-RL tasks in the windy Gridworld environments.

the agent must navigate around impassable terrain (e.g., walls and lava) and interact with specific objects (e.g., keys and doors). Winds are introduced in the passages between lava as unobserved confounders, affecting the agent's movements. For each task, interactive objects are assigned colors from a set of six; one color is designated as the unique target, while the remaining three serve as distractions. The source domain uses the palette {red, green, blue, purple}, while the target domain extends it with two additional colors, {yellow, gray}. We evaluate our approach on three meta-RL tasks: Pick-Up-Key (Experiment 1), Go-To-Door (Experiment 2), and Go-To-Goal (Experiment 3). Each environment contains four tasks in the source domain and two tasks in the target domain.

We assess the performance of algorithms by their ability to adapt to target tasks, specifically, quantified by the accumulated reward obtained during adaptation. For all baselines, the meta-policy is adapted to the target task using Proximal Policy Optimization (PPO) (Schulman et al., 2017a). Our method is compared to three baselines: (a) PPO: random

initialization of meta-policy parameters; (b) MAML: training the meta-policy on demonstrator data using MAML; (c) $RL^2$ (Duan et al., 2017): training the meta-policy on demonstrator data using $RL^2$, and (d) PRETRAINED-PPO: pretraining the meta-policy on demonstrator data. Additional details about the experimental setups are provided in Sec. D. This section focuses on the evaluation of Causal MAML in Alg. 1. However, the proposed counterfactual augmentation could be applied to other popular meta-RL algorithms to mitigate the confounding bias. We refer readers to Sec. B for additional experiments.

The policy model for the actor-critic network consists of a two-headed multilayer perceptron (MLP). Both the actor and critic heads share a fully connected layer with 64 units, and each head features a single hidden layer MLP with 64 hidden units. During the meta-training stage, we train the model for 300 iterations. In

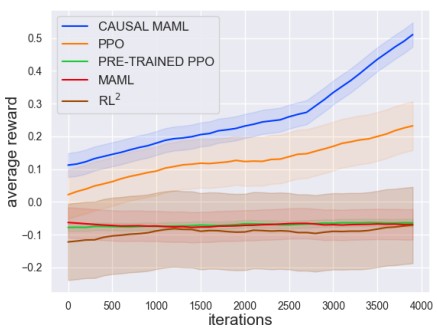

*(a)* Go-To-Door

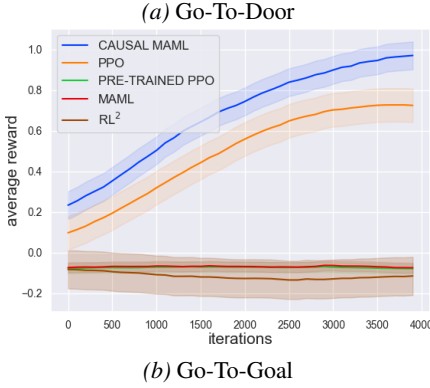

*(b)* Go-To-Goal

*Figure 6.* Cumulative returns comparing PPO from scratch, PRETRAINED PPO, standard MAML, and proposed CAUSAL-MAML.

the adaptation stage, we select five tasks from the target domain, train for $4,000$ iterations, and calculate the average accumulated reward across the tasks. Each iteration uses 512 frames from the environments.

**Experiment 1.** In the first experiment, the agent is trained to navigate in a $15 \times 9$ grid and to find the key of the target color. Details of this meta-RL task have been described in Fig. 1a. Keys are uniformly generated within the subgrid $\{(c, r) \mid 7 \le c \le 13, 4 \le r \le 7\}$. The wind distribution in the passages between lava is $0.1, 0.35, 0.1, 0.35, 0.1$ for rightward, downward, leftward, upward, and staying in place, respectively. In other cells, the distribution is $0.01, 0.01, 0.01, 0.01, 0.96$, indicating negligible wind effects. If the agent enters lava, a negative reward is received, while approaching the target key yields a positive reward. Simulation results in Fig. 1b suggest that confounding robust Meta-RL

adapts more quickly and exhibits lower variance during adaptation compared to PPO. MAML, PRETRAINED-PPO, and $RL^2$ fail to learn from confounded data.

**Experiment 2.** In the second experiment, the agent is required to pick up the target color key and open the corresponding door in a $15 \times 9$ grid. The environment is illustrated in Fig. 5a. Key locations are uniformly generated from the set $\{(7, 2), (9, 1), (9, 4), (9, 7)\}$, and door locations are uniformly generated from the set $\{(13, 1), (13, 3), (13, 5), (13, 7)\}$. The wind distribution in the lava passage and other cells is identical to the description in Experiment 1. Entering lava produces a negative reward. Before obtaining the target key, approaching it yields a positive reward; after acquiring the target key, approaching the corresponding door provides a positive reward. As shown in Fig. 6a, our proposed CAUSAL-MAML also adapts more quickly than PPO with lower variance, while MAML, PRETRAINED-PPO, and $RL^2$ are hurt by confounded data and fail to discover the correct path.

**Experiment 3.** In the third experiment, the agent should pick up the target color key, open the corresponding door, and reach the goal in a $18 \times 9$ grid. An illustration of the environment is provided in Fig. 5b. Key locations are uniformly generated from the set $\{(7, 2), (9, 1), (9, 4), (9, 7)\}$, door locations are uniformly generated from the set $\{(13, 1), (13, 3), (13, 5), (13, 7)\}$, and the goal are generated within the subgrid $\{(c, r) \mid 13 \le c \le 16, ; 6 \le r \le 7\}$. The wind distribution is the same as that in Experiment 1. Before obtaining the target key, approaching it yields a positive reward; after acquiring the target key, approaching the goal provides a positive reward. Fig. 6b indicates that our proposed CAUSAL-MAML outperforms PPO and MAML in terms of adaptation speed and variance reduction. MAML is able to identify the correct path, while PRETRAINED-PPO and $RL^2$ ARE unable to converge.

## 5. Conclusion

This paper examines a critical limitation in existing meta-RL algorithms: their susceptibility to unmeasured confounding in observational data. To overcome this challenge, we introduce a novel data augmentation procedure for meta-RL. Our framework provides a principled method to identify and reason about potential counterfactual environments consistent with the observed data by leveraging causal inference techniques. We then train a meta-policy by interacting with the generated counterfactual environments, allowing the agent to learn from unbiased experiences and acquire more robust, generalizable skills. Our theoretical analysis establishes the convergence guarantees of our counterfactual bootstrap approach. Future research could explore extending this framework to high-dimensional environments with more complex, multimodal action-state distributions.

## Acknowledgment

This work was supported by the Natural Science Foundation (NSF) Grant (CNS-2221875).

## Impact Statement

This paper presents a novel augmentation technique, counterfactual bootstrap, that generates new counterfactual trajectories for training a meta-policy by leveraging confounded observational data.

Our proposed counterfactual bootstrap can be integrated into standard meta RL algorithms to improve their robustness against confounding biases. Such capabilities could be applied to a range of domains, such as healthcare and medicine, Economics and Public Policy, as well as robotics and autonomous vehicles, where randomized experimentation is often costly, risky, or infeasible.

The augmentation technique constructs a set of plausible simulated environments consistent with the confounding biased observation data. The procedure offers several advantages: (i) it can substantially reduce the consumption of social and economic resources when estimating the causal effects of interventions, such as the impact of a new medication or public policy. (ii) it improves safety by reducing the need for direct interaction with real-world environments during early-stage training. For example, training an autonomous driving agent from scratch through real-world interaction is impractical and unsafe. Directly leveraging expert driver demonstrations may fail, since human drivers may attend to cues (e.g., subtle visual or contextual signals) that are not captured by sensors. Reconstructing counterfactual trajectories provides a more feasible and safer alternative in such extreme cases.

It is important to emphasize that the meta-policy trained using counterfactual data is designed to capture generalized behavioral patterns across environments, rather than a deployable policy. As such, an adaptation or specialization procedure is required to tailor the meta-policy to a specific environment or deployment context. We therefore encourage future research to focus on effective adaptation mechanisms that safely and reliably specialize meta-policies for concrete real-world settings.

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

# A. Related Works

## A.1. Causal Reinforcement Learning

Causal RL is a burgeoning research direction that introduces causality into the field of RL. Some works incorporate causal structure to address distribution shift and heterogeneous environments. (de Haan et al., 2019) show that more information can hurt behavioral cloning by obscuring causal structure, and propose targeted interventions to recover the correct causal model, though only in-distribution. (Bellot et al., 2023) study heterogeneous data transfer for sequential decision-making but assume identifiable sources without within-source confounding. (Li et al., 2023) leverages implicit causal structure to improve exploration, relying on online interaction during training. (Balazadeh et al., 2024) incorporate demonstrations via Bayesian inference over latent variables, yet do not consider the meta-learning setting. (Mu et al., 2022) proposes a meta-reinforcement learning framework that learns a generalized task context for effect modifiers across environments but does not consider real confounding environments. (Li et al., 2026) extends the Bellman equation to a causal Bellman equation that lower-bounds expected return from confounded off-policy data and uses it to build Causal-DQN. Our work is, to our knowledge, the first to combine partial identification with meta-RL to handle unmeasured confounding across heterogeneous source tasks.

## A.2. Posterior-based Reinforcement Learning

To address task heterogeneity under uncertainty, another major line of work focuses on posterior-based reinforcement learning. A large body of meta-RL research addresses task heterogeneity by maintaining probabilistic beliefs over latent environments. Posterior sampling methods such as (Osband et al., 2013) model uncertainty over MDP dynamics to guide exploration. Bayesian meta-learning approaches further learn task representations enabling rapid adaptation across environments, including the works of (Yoon et al., 2018; Grant et al., 2018). Task inference methods, such as those proposed by (Rakelly et al., 2019; Zintgraf et al., 2021), infer latent context variables online to adapt policies. Despite their differences, these approaches share a common assumption: the environment is identifiable and can be captured by a correctly specified probabilistic model. Consequently, they may fail under unmeasured confounding or model misspecification.

## A.3. Meta Reinforcement Learning

Meta-reinforcement learning (meta-RL) seeks to train agents that can rapidly adapt to new tasks drawn from a task distribution by leveraging experience across related tasks. (Duan et al., 2016) and (Wang et al., 2016) frame meta-RL as "learning to learn," training a recurrent network across a distribution of MDPs so that adaptation emerges entirely within the RNN's hidden state. (Finn et al., 2017) instead learn an initialization from which one or a few policy-gradient steps suffice to solve a new task. SNAIL (Mishra et al., 2018) combines temporal convolutions with soft attention to aggregate past experience more flexibly. PEARL (Rakelly et al., 2019) disentangles inference from control via probabilistic latent task embeddings learned from off-policy data, while VariBAD (Zintgraf et al., 2021) casts the problem as approximate Bayes-optimal behavior in a BAMDP by maintaining a learned belief over task variables. However, these methods all rest on the assumption that the environment is identifiable, and are prone to failure in the presence of confounding.

## A.4. Safe Reinforcement Learning

Safe reinforcement learning aims to maximize task reward while satisfying safety constraints during both training and deployment. (Achiam et al., 2017) introduce Constrained Policy Optimization (CPO), a trust-region method that enforces safety constraints throughout training on high-dimensional robotic locomotion tasks. To avoid the computational overhead of trust-region projections, a parallel line of research adopts primal-dual Lagrangian approaches that penalize the reward with adaptive multipliers on constraint violations (Tessler et al., 2018). (Liu et al., 2022) propose Constrained Variational Policy Optimization (CVPO), which reformulates safe RL as probabilistic inference via an Expectation-Maximization scheme. Conditioned Constrained Policy Optimization (CCPO) (Yao et al., 2023) is introduced to learn versatile safe policies capable of zero-shot adaptation to varying constraint thresholds at deployment without retraining. Although these methods incorporate safety constraints during training, they largely overlook the influence of confounding factors, which makes it difficult to learn feasible policies in confounded environments.

# B. Generalization to Other Meta-RL Algorithms

The proposed counterfactual bootstrap procedure is general and can thus be naturally incorporated into other meta-RL algorithms to mitigate the influence of confounding bias. The remainder of this section is organized as follows. Sec. B.1 describes how to incorporate the counterfactual bootstrap into two popular RL methods for meta-learning tasks, including pretraining with PPO (Schulman et al., 2017a), RL$^2$ (Duan et al., 2017) and Decision Pretrained Transformer (DPT) (Lee et al., 2023). Sec. B.2 provides additional experimental results demonstrating the consistent performance improvement of the augmented causal mete-RL methods compared to their non-causal counterparts.

## B.1. Additional Causal Meta-RL Methods

Pretraining with PPO, RL$^2$ and DPT are three baseline methods evaluated in Sec. 4. Simulation results indicate that they fail to learn effectively from confounded observational data. In this section, we will show how the counterfactual bootstrap can be applied to augment these algorithms to mitigate confounding bias.

To make this argument more concrete, we note that the counterfactual bootstrap can be leveraged to mitigate confounding bias arising from any finite set of observational data. Given any observed dataset, the learner could sample a counterfactual environment $\widehat{\mathcal{M}}$ from the posterior $\widehat{\rho}(\mathcal{M} \mid \mathcal{D}_{\text{obs}})$, interact with the sampled instance $\widehat{\mathcal{M}}$, and collect subsequent experimental data $\widehat{\mathcal{D}}_{\text{exp}}$. The meta-RL algorithm could then proceed accordingly by replacing the original offline observations $\mathcal{D}_{\text{obs}}$ with the sampled experimental data $\widehat{\mathcal{D}}_{\text{exp}}$ collected from the counterfactual bootstrap. For the CMDP environments considered in this work, we could approximate the system dynamics of the posterior CMDP model by drawing parameters of the transition and reward functions within the feasible region boundary specified in Eqs. (5) to (8).

Details of the augmented PRETRAINED-PPO, RL$^2$, and DPT are summarized in Algs. 2 and 3. Specifically, Alg. 2 presents the CAUSAL-PRETRAINED-PPO method. Similar to Alg. 1, we first sample a batch of CMDP tasks from the distribution $\rho(\mathcal{M})$. For each task in the batch, we collect observational data and construct an approximate environment $\widehat{\mathcal{M}}_i$. The agent then interacts with this reconstructed environment, computes the PPO loss using the resulting on-policy trajectories, and updates its parameters accordingly.

Alg. 3 details the CAUSAL-RL$^2$ method. RL$^2$ employs a recurrent neural network (RNN) architecture to enable within-task adaptation. Before interacting with task $\mathcal{M}_i$, we initialize the RNN hidden state $h$. As in CAUSAL-PRETRAINED-PPO, we construct approximate environments from the observational data. During trajectory sampling, the hidden state $h$ is updated at each step. Finally, we compute the gradient of the RL$^2$ loss and update the agent parameters.

CAUSAL-DPT follows the same training pipeline of PRETRAINED-PPO, as presented in Alg. 2. However, CAUSAL-DPT additionally incorporates information from the previous timestep to predict the probability of the current action. Specifically, both the actor and critic networks take the tuple $(s_{t-1}, a_{t-1}, r_{t-1}, s_t)$ as input to predict the next action and estimate the state value.

## B.2. Additional Empirical Evaluations

We evaluate the augmented PRETRAINED-PPO, RL$^2$ and DPT using the counterfactual bootstrap across the Pick-Up-Key, Go-To-Door, and Go-To-Goal environments and compare their performance with their standard counterparts and with vanilla PPO. The return curves are shown in Fig. 7. Each row in Fig. 7 corresponds to a specific Meta-RL method, while each column represents a particular evaluation environment. Overall, we found that the counterfactual bootstrap consistently improves the performance of the baseline meta-RL algorithms. Augmented Causal Meta-RL methods can learn an effective policy initialization from confounded observations that generalizes well to downstream RL tasks.

As shown in (Fig. 7a–Fig. 7c), CAUSAL-MAML consistently achieves higher returns than both standard MAML and PPO, demonstrating its ability to effectively learn under confounded observation biases where standard MAML fails to adapt. The performance gap between CAUSAL-MAML and standard MAML is obviously present in the Pick-Up-Key task (Fig. 7a), while more moderate but consistent improvements are observed in the Go-To-Door and Go-To-Goal environments (Fig. 7b, Fig. 7c). Similarly, results in (Fig. 7d–Fig. 7f) indicate that CAUSAL-PRETRAINED-PPO substantially outperforms both PRETRAINED-PPO and vanilla PPO across all three tasks. This highlights that incorporating causal counterfactual augmentation enables the agent to overcome confounding effects that otherwise prevent effective transfer and adaptation. Fig. 7d shows a significant improvement of CAUSAL-PRETRAINED-PPO in Pick-Up-Key task and consistent improvements are observed in the Go-To-Door and Go-To-Goal environments (e, f). (Fig. 7g–Fig. 7i) show

---

**Algorithm 2:** CAUSAL-PRETRAINED-PPO

---

1   **Require:** Initial parameter $\theta$, an approximate prior over CMDPs $\widehat{\rho}(\mathcal{M})$
2   **while** not done **do**
3      Nature samples a batch of CMDP tasks $\mathcal{B} = \{\mathcal{M}_i\}_{i=1}^{B}$ from distribution $\rho(\mathcal{M})$
4      **for** all task $\mathcal{M}_i \in \mathcal{B}$ **do**
5          Sample observation trajectories $\mathcal{D}_{\text{obs}}^{i}$ in environment $\mathcal{M}_i$
6          Sample a new environment $\widehat{\mathcal{M}}_i$ from the posterior $\widehat{\rho}(\mathcal{M} \mid \mathcal{D}_{\text{obs}}^{i})$
7          Sample experimental trajectories $\widehat{\mathcal{D}}_{\text{exp}}^{i}$ using agent policy $\pi(\cdot \mid \cdot; \theta)$ in environment $\widehat{\mathcal{M}}_i$
8          Compute gradient $\widehat{\nabla}_\theta J_i(\theta, \widehat{\mathcal{D}}_{\text{exp}}^{i})$ using dataset $\widehat{\mathcal{D}}_{\text{exp}}^{i}$
9          Update parameter $\theta \leftarrow \theta + \alpha \widehat{\nabla} J_i(\theta, \widehat{\mathcal{D}}_{\text{exp}}^{i})$
10      **end**
11   **end**

---

**Algorithm 3:** CAUSAL-RL$^2$

---

1   **Require:** Initial parameter $\theta$, an approximate prior over CMDPs $\widehat{\rho}(\mathcal{M})$
2   **while** not done **do**
3      Nature samples a batch of CMDP tasks $\mathcal{B} = \{\mathcal{M}_i\}_{i=1}^{B}$ from distribution $\rho(\mathcal{M})$
4      **for** all task $\mathcal{M}_i \in \mathcal{B}$ **do**
5          Initialize RNN hidden state $h$
6          Sample observation trajectories $\mathcal{D}_{\text{obs}}^{i}$ in environment $\mathcal{M}_i$
7          Sample a new environment $\widehat{\mathcal{M}}_i$ from the posterior $\widehat{\rho}(\mathcal{M} \mid \mathcal{D}_{\text{obs}}^{i})$
8          Sample experimental trajectories $\widehat{\mathcal{D}}_{\text{exp}}^{i}$ and update RNN hidden state $h$ using agent policy $\pi(\cdot \mid \cdot; \theta)$ in environment $\widehat{\mathcal{M}}_i$
9          Compute gradient $\widehat{\nabla}_\theta J_i(\theta, \widehat{\mathcal{D}}_{\text{exp}}^{i})$ using dataset $\widehat{\mathcal{D}}_{\text{exp}}^{i}$
10          Update parameter $\theta \leftarrow \theta + \alpha \widehat{\nabla} J_i(\theta, \widehat{\mathcal{D}}_{\text{exp}}^{i})$
11      **end**
12   **end**

---

that CAUSAL-RL$^2$ also benefits significantly from the proposed causal intervention, achieving faster convergence and higher returns compared to its standard RL$^2$ counterpart. CAUSAL-RL$^2$ achieves substantially higher returns in Pick-Up-Key (g) and Go-To-Door (h), with consistent improvements in Go-To-Goal (i). (Fig. 7j–Fig. 7l) demonstrate CAUSAL-DPT consistently outperforms both PRETRAINED-PPO and standard DPT in Pick-Up-Key and Go-To-Door environments, and achieves higher returns in the later stages of training in Go-To-Goal environment. Overall, these results demonstrate the robustness and generality of our augmentation technique across different meta-RL paradigms and environments.

### B.3. Comparison Among Causal Meta-RL Algorithms

Our simulation results demonstrate that the counterfactual bootstrap can be effectively applied across various meta-RL algorithms. In all experiments, the causally augmented meta-learners consistently outperformed their non-causal counterparts. However, in this paper, we do not make a definitive claim about which causal meta-RL algorithm is superior. Our analysis, along with previous research, suggests that the optimal choice of causal meta-learner depends on the specific learning tasks and their parameters.

To clarify our argument, we note that pre-training methods can sometimes outperform meta-learning approaches, as indicated in the literature (Zhao et al., 2022; Gao & Sener, 2020). This observation is relevant for all meta-RL approaches that utilize expected risk minimization, including model-agnostic meta-learning (MAML) and Bayesian MAML. There is no universal agreement on whether pre-training or meta-learning is superior; the choice ultimately depends on the diversity of the available data.

However, our goal in this paper is not to resolve the debate between pre-training and meta-learning. Instead, we propose a

general data augmentation technique that can enhance standard few-shot reinforcement learning (RL) algorithms, making them more robust against confounding biases. In this work, we focus on MAML as our primary target because it is the most widely used and is regarded as the canonical meta-RL algorithm. Nonetheless, our learning strategy can be easily integrated into other few-shot learning algorithms, consistently improving their robustness in the presence of unobserved confounding.

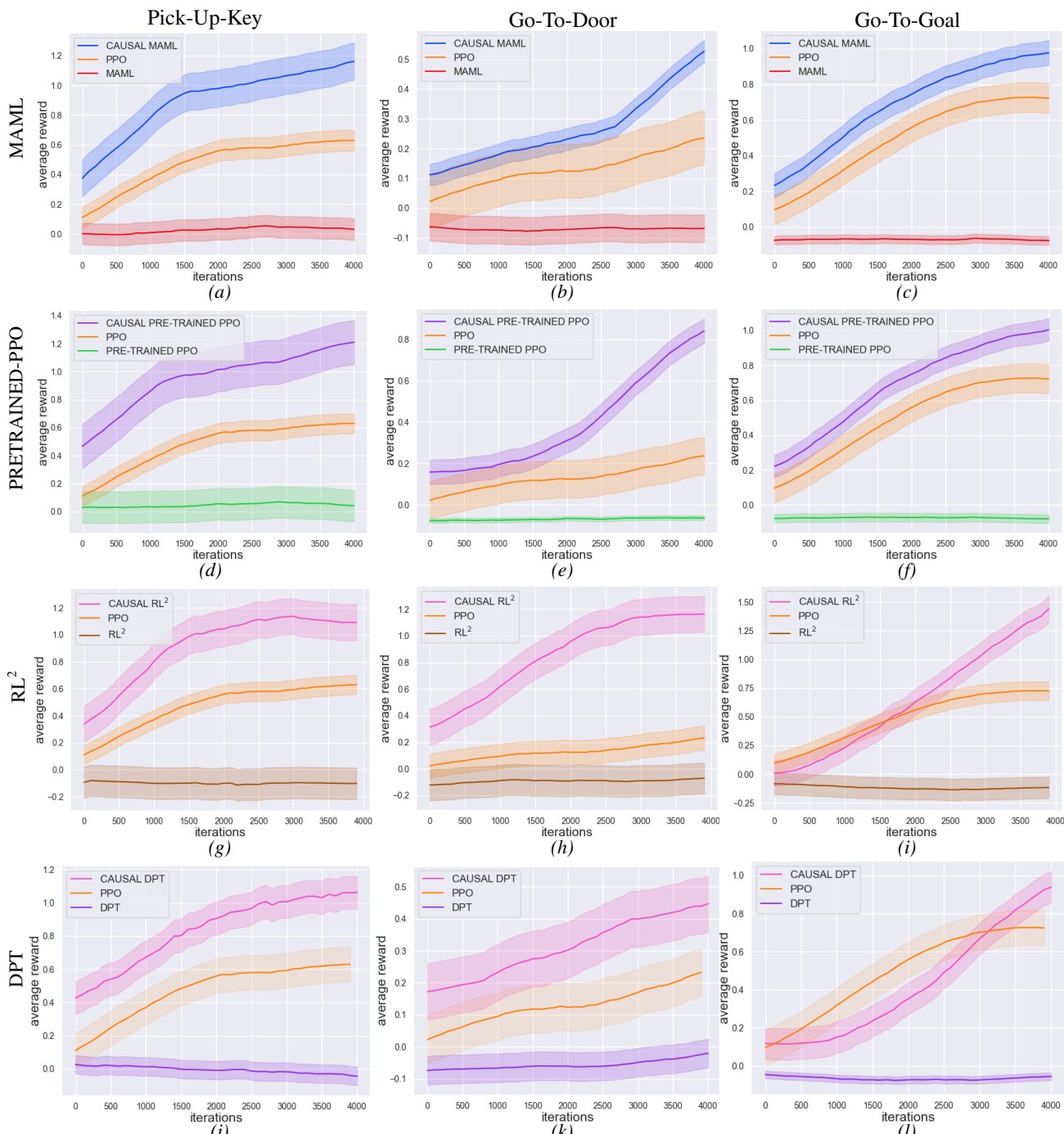

*Figure 7.* Returns on MiniGrid environments comparing all baselines across Pick-Up-Key, Go-To-Door, and Go-To-Goal tasks. Each plot compares the augmentation-based Meta-RL method, standard PPO, and standard Meta-RL method. The Meta-RL methods include: (a - c) MAML; (d - f) PRETRAINED-PPO; (g - i) RL$^2$; and (j - l) DPT.

# C. Ablation Study

In this section, we demonstrate the robustness and reliability of our proposed counterfactual bootstrap under varying levels of confounding strength. We first show how Manski bounds can be tightened by incorporating domain-specific inductive biases. We then conduct ablation studies for CAUSAL-MAML in weak confounding environments, demonstrating that our augmentation strategy does not degrade as the confounding strength varies.

In Eqs. (5) to (8), we present the assumption-free natural Manski bounds for the transition probability $\mathcal{T}_i(s, x, s')$ and reward function $\mathcal{R}_i(s, x)$. These bounds can be further tightened by incorporating domain-specific inductive biases. The transition probability can be decomposed into aligned and misaligned components according to the demonstrator's behavior:

$$\mathcal{T}_i(s, x, s') = P(s' \mid s, do(x)) = \underbrace{\sum_{u:f_X(s,u)=x} P(s' \mid s, x, u)P(u \mid s)}_{\text{aligned: demonstrator chose } x} + \underbrace{\sum_{u:f_X(s,u)\neq x} P(s' \mid s, x, u)P(u \mid s)}_{\text{misaligned: demonstrator chose } \neq x} \quad (12)$$

The aligned term corresponds to latent states in which the demonstrator naturally selects action $x$. Since these state–action–outcome triples are observed in the data, this component marginalizes to $P(s' \mid s, x)P(x \mid s)$ and is therefore fully identifiable. In contrast, the misaligned term involves the counterfactual quantity $P(s' \mid s, x, u)$ for latent states where the demonstrator would not have chosen action $x$—precisely the states that remain unobserved under intervention $x$. Because $P(s' \mid s, x, u) \in [0, 1]$, this term is naturally bounded by $[0, P(\neg x \mid s)]$, which yields the bounds presented in Eqs. (5) and (6).

Under weak confounding, we may assume that the counterfactual quantity satisfies $P(s' \mid s, x, u) \in [P(s' \mid s, x) - \alpha, P(s' \mid s, x) + \beta]$ for constants $\alpha, \beta \geq 0$. This assumption confines the transition probability to a narrower interval:

$$\mathcal{T}_i(s, x, s') \in [P(s'|x, s) - \alpha P(\neg x \mid s), P(s'|x, s) + \beta P(\neg x \mid s)] \quad (13)$$

Similarly, under weak confounding, the Manski bounds for the reward function $\mathcal{R}_i(s, x)$ become

$$\mathcal{R}_i(s, x) \in [r(s, x) - \alpha P(\neg x \mid s), r(s, x) + \beta P(\neg x \mid s)] \quad (14)$$

The degree of confounding in our MiniGrid environments is controlled by the wind distribution, which varies across grid positions. In the standard setting, the probability of no wind in lava passages is 0.1, corresponding to a strong confounding regime. For the ablation study, we increase the probability of no wind in lava passages to 0.9, thereby creating a weak confounding setting. The training returns across the Pick-Up-Key, Go-To-Door, and Go-To-Goal tasks are shown in Fig. 8. The results show that as confounding weakens, the performance gap between CAUSAL-MAML and standard MAML correspondingly narrows, with CAUSAL-MAML achieving performance comparable to MAML. This finding confirms that our causal augmentation does not degrade performance when confounding is negligible; instead, the method gracefully recovers standard meta-RL behavior without introducing unnecessary conservatism.

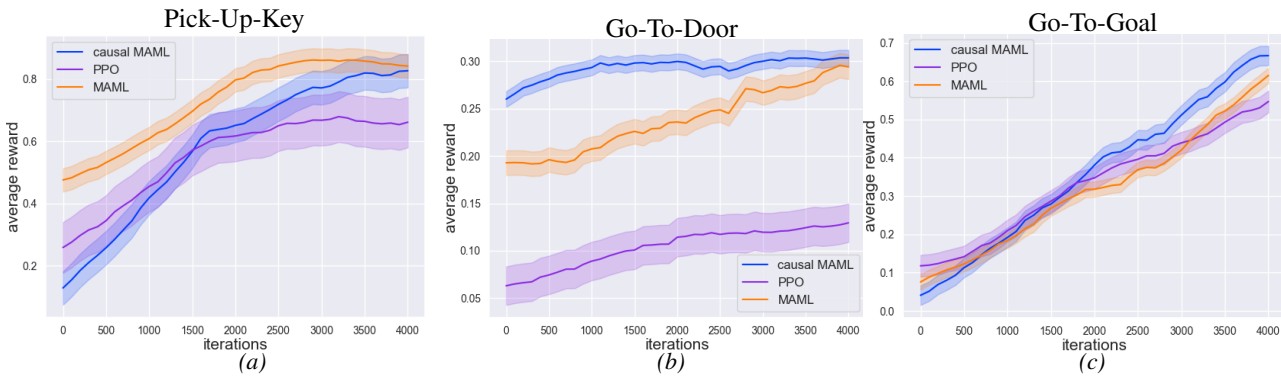

*Figure 8.* Returns of CAUSAL-MAML on MiniGrid environments across the Pick-Up-Key, Go-To-Door, and Go-To-Goal tasks under weak confounding settings.

# D. Experimental Setups

This section provides additional details about the experimental setups. All meta-RL tasks build on the Windy Gridworlds (Li et al., 2025a; Zhang & Bareinboim, 2025), which is adapted from the Minigrid environment (Chevalier-Boisvert et al., 2023). In these environments, the agent must navigate around impassable terrain (e.g., walls and lava) and interact with specific objects (e.g., keys and doors). Winds are introduced in the passages between lava as unobserved confounders, affecting the agent's movements. For each task, interactive objects are assigned colors from a set of six; one color is designated as the unique target, while the remaining three serve as distractions. The source domain uses the palette {red, green, blue, purple}, while the target domain extends it with two additional colors, {yellow, gray}. We evaluate our approach on three meta-RL tasks: Pick-Up-Key (Experiment 1), Go-To-Door (Experiment 2), and Go-To-Goal (Experiment 3). Each environment comprises four source-domain tasks and two target-domain tasks.

We assess the performance of algorithms by their ability to adapt to target tasks, specifically, quantified by the accumulated reward obtained during adaptation. Experiments in the main paper (Sec. 4) evaluate Causal MAML described in Alg. 1. Additionally, we apply the counterfactual bootstrap to PPO and $RL^2$ and provide their implementations in Sec. B.1. Additional simulation results of these variants of causal meta-RL algorithms are provided in Sec. B.2. These algorithms are compared to three baselines: (a) PPO: random initialization of meta-policy parameters; (b) MAML: training the meta-policy on demonstrator data using MAML; (c) $RL^2$ (Duan et al., 2017): training the meta-policy on demonstrator data using $RL^2$, and (d) PRETRAINED-PPO: pretraining the meta-policy on demonstrator data.

For all experiments, the policy model for the actor-critic network consists of a two-headed multilayer perceptron (MLP). Both the actor and critic heads share a fully connected layer with $64$ units, and each head has a single hidden-layer MLP with $64$ hidden units. During the meta-training stage, we train the model for $300$ iterations. In the adaptation stage, we select five tasks from the target domain, train for $4,000$ iterations, and calculate the average accumulated reward across the tasks. Each iteration uses $512$ frames from the environments. Details of the hyperparameters used in training causal meta-RL algorithms are provided in Tables 1 to 3.

*Table 1.* Hyperparameter of CAUSAL-MAML.

| Hyperparameter | Hyperparameter Values |
|---|---|
| Outer Loop Learning Rate | $3e-4$ |
| Inner Loop Learning Rate | $1e-4$ |
| Frames Per Batch | 512 |
| Batch Size | 128 |
| Number Of Meta-Learning Episode | 300 |
| Number Of Adaption Episode | 4000 |
| Environment Max Steps | 512 |
| Outer Loop Optimizer | Adam |
| Inner Loop Optimizer | SGD |
| Activations | Tanh |
| Actor MLP Hidden Nodes | [32, 32, 32] |
| Value MLP Hidden Nodes | [32, 32, 32] |
| $\gamma$ | 0.99 |
| $\lambda$ | 0.95 |
| Entropy Coefficient | $1e-4$ |
| Critic Coefficient | 1.0 |

*Table 2.* Hyperparameter of CAUSAL-PRETRAINED-PPO.

| Hyperparameter | Hyperparameter Values |
|---|---|
| Learning Rate | $3e-4$ |
| Frames Per Batch | 512 |
| Batch Size | 128 |
| Number Of Meta-Learning Episode | 300 |
| Number Of Adaption Episode | 4000 |
| Environment Max Steps | 512 |
| Optimizer | Adam |
| Activations | Tanh |
| Actor MLP Hidden Nodes | [32, 32, 32] |
| Value MLP Hidden Nodes | [32, 32, 32] |
| Clip Ratio | 0.2 |
| $\gamma$ | 0.99 |
| $\lambda$ | 0.95 |
| Entropy Coefficient | $1e-4$ |
| Critic Coefficient | 1.0 |

*Table 3.* Hyperparameter of CAUSAL-RL[2].

| Hyperparameter | Hyperparameter Values |
|---|---|
| Learning Rate | $3e-4$ |
| Batch Epochs | 30 |
| Number Of Meta-Learning Episode | 300 |
| Number Of Adaption Episode | 4000 |
| Environment Max Steps | 128 |
| Optimizer | Adam |
| Activations | Tanh |
| RNN Type | LSTM |
| RNN Size | 64 |
| Length of BPTT | 32 |
| Actor MLP Hidden Nodes | [32, 32] |
| Value MLP Hidden Nodes | [32, 32] |
| Clip Ratio | 0.2 |
| $\gamma$ | 0.99 |
| $\lambda$ | 0.95 |
| Entropy Coefficient | $1e-4$ |
| Critic Coefficient | 1.0 |

# E. Proof Details

In this section, we provide the detailed proof of the convergence of our CAUSAL-MAML method. We begin by presenting two lemmas that serve as the foundation of the proof. We then outline the proof process for these lemmas, followed by the proof of the main theorem.

## E.1. Details of Convergence proof

Establishing the Lipschitz property of the meta-objective function requires information from the task-specific objective functions $J_i(\theta)$, along with their gradient $\nabla_\theta J_i(\theta)$ and Hessian matrix $\nabla_\theta^2 J_i(\theta)$. Referring to the results in (Shen et al., 2019), we state the following lemmas on the Lipschitz property of the accumulated reward function $J_i(\theta)$.

**Lemma 1.** *Define $R = \max(|a|, |b|)$. Suppose Assumptions 1 and 2 hold, we have:*

(i) *$J_i(\theta)$ is smooth with parameters $\eta_G := \frac{RG}{(1-\gamma)^2}$; that is, for any parameter $\theta \in \mathbb{R}^d$, we have $\|\nabla_\theta J_i(\theta)\| \le \eta_G$.*

(ii) *$\nabla_\theta J_i(\theta)$ is smooth with parameters $\eta_H := \frac{(H+1)RG^2+RL}{(1-\gamma)^2}$; that is, for any parameter $\theta \in \mathbb{R}^d$, we have $\|\nabla_\theta^2 J_i(\theta)\| \le \eta_H$.*

(iii) *$\nabla_\theta^2 J_i(\theta)$ is smooth with parameters $\eta_\rho := \frac{2(H+1)RGL+RK}{(1-\gamma)^2}$; that is, for any parameter $\theta_1, \theta_2 \in \mathbb{R}^d$, we have $\|\nabla_\theta^2 J_i(\theta_1) - \nabla_\theta^2 J_i(\theta_2)\| \le \eta_\rho \|\theta_1 - \theta_2\|$.*

Lemma 1 demonstrates that the Lipschitz parameters of the task-specific objective function $J_i(\theta)$, its gradient $\nabla_\theta J_i(\theta)$, and its Hessian $\nabla_\theta^2 J_i(\theta)$ are $\eta_G, \eta_H, \eta_\rho$, respectively. Based on the result in Lemma 1, we can now demonstrate the Lipschitz property of the meta-objective function. The stochastic gradient of the meta-objective function is defined as follows:

$$\widehat{\nabla}_\theta F(\theta) = \frac{1}{|\mathcal{B}|} \sum_{i \in \mathcal{B}} \left( \left( I + \alpha \widehat{\nabla}_\theta^2 J_i(\theta, \widehat{\mathcal{D}}_{\exp,\text{in}}^i) \right) \widehat{\nabla}_\theta J_i \left( \theta_i, \widehat{\mathcal{D}}_{\exp,\text{o}}^i \right) \right.$$
$$\left. + \widehat{J}_i \left( \theta_i, \widehat{\mathcal{D}}_{\exp,\text{o}}^i \right) \sum_{\tau \in \widehat{\mathcal{D}}_{\exp,\text{in}}^i} \sum_{t=0}^{H} \nabla_\theta \log \pi(x_t \mid s_t; \theta) \right). \tag{15}$$

Referring to the result in (Fallah et al., 2021), we state the following conclusion on Lipschitz property of meta-objective function $F(\theta)$.

**Lemma 2.** *Consider the meta-objective function defined in Eq. (4) for the case that $\alpha \in (0, \frac{1}{\eta_H}]$. Suppose Assumptions 1 and 2 are satisfied. Then, we have:*

(i) *$\widehat{\nabla}_\theta F(\theta)$ is bounded by parameter $L_G = \frac{2RG}{(1-\gamma)^2} + \frac{D(H+1)RG}{1-\gamma}$; that is, for any parameter $\theta$, any task subset $\mathcal{B}$, and any experimental trajectory batch $\widehat{\mathcal{D}}_{exp}^i$, we have $\|\widehat{\nabla}_\theta F(\theta)\| \le L_G$.*

(ii) *$\widehat{\nabla}_\theta F(\theta)$ is smooth with parameter $L_H = 4\eta_H + \alpha\eta_G\eta_\rho + D(H+1)R(\frac{L}{1-\gamma} + \frac{2G^2}{(1-\gamma^2)})$; that is, for any parameter $\theta$, any task subset $\mathcal{B}$, and any experimental trajectory batch $\widehat{\mathcal{D}}_{exp}^i$, we have $\|\widehat{\nabla}_\theta^2 F(\theta)\| \le L_H$.*

Lemma 2 illustrates the upper bound and the Lipschitz parameter of the stochastic gradient $\widehat{\nabla}_\theta F(\theta)$.

## E.2. Proof of Lemma 1

In this section, we show the proof details of Lemma 1.

**Proof of (i):**

First, we note that

$$\left\|\sum_{t=0}^{H} \nabla_\theta \log \pi(x_t \mid s_t; \theta)\Psi_t\right\| \le \sum_{t=0}^{H} \|\nabla_\theta \log \pi(x_t \mid s_t; \theta)\| |\Psi_t|$$

$$\le \sum_{t=0}^{H} |\Psi_t| G.$$

The accumulated reward is

$$|\Psi_t| = \left|\sum_{t'=t}^{H} \gamma^t R_i(s_{t'}, x_{t'})\right|$$

$$\le R \sum_{t'=t}^{H} \gamma^{t'}$$

$$\le \frac{R\gamma^{t'}}{1 - \gamma}.$$

Consequently, we have

$$\left\|\sum_{t=0}^{H} \nabla_\theta \log \pi(x_t \mid s_t; \theta)\Psi_t\right\| \le RG \sum_{t=0}^{H} \frac{\gamma^{t'}}{1 - \gamma}$$

$$\le \frac{RG}{(1 - \gamma)^2}.$$

**Proof of (ii):**

Note that

$$\left\|(\sum_{t=0}^{H} \nabla_\theta \log \pi(x_t \mid s_t; \theta)\Psi_t)\nabla_\theta \log q_i(\tau; \theta)^\mathsf{T} + \sum_{t=0}^{H} \nabla_\theta^2 \log \pi(x_t \mid s_t; \theta)\Psi_t\right\|$$

$$\le \left\|\sum_{t=0}^{H} \nabla_\theta \log \pi(x_t \mid s_t; \theta)\Psi_t)\right\| \|\nabla_\theta \log q_i(\tau; \theta)\| + \left\|\sum_{t=0}^{H} \nabla_\theta^2 \log \pi(x_t \mid s_t; \theta)\Psi_t\right\|.$$

First, we consider the bound on $\|\nabla_\theta \log q_i(\tau; \theta)\|$:

$$\|\nabla_\theta \log q_i(\tau; \theta)\| = \sum_{t=0}^{H} \|\nabla_\theta \log \pi(x_t \mid s_t; \theta)\|$$

$$\le (H + 1)G$$

According to the result in Lemma 1(i),

$$\left\|\sum_{t=0}^{H} \nabla_\theta \log \pi(x_t \mid s_t; \theta)\Psi_t\right\| \le \frac{RG}{(1 - \gamma)^2}.$$

Then, we consider the bound on $\|\sum_{t=0}^{H} \nabla_\theta^2 \log \pi(x_t \mid s_t; \theta)\Psi_t\|$:

$$\left\|\sum_{t=0}^{H} \nabla_\theta^2 \log \pi(x_t \mid s_t; \theta)\Psi_t\right\| \le \sum_{t=0}^{H} \|\nabla_\theta^2 \log \pi(x_t \mid s_t; \theta)\| |\Psi_t|$$

$$\le RL \sum_{t=0}^{H} \frac{\gamma^{t'}}{1 - \gamma}$$

$$\le \frac{LR}{(1 - \gamma)^2}.$$

Consequently, we have

$$
\left\| (\sum_{t=0}^{H} \nabla_\theta \log \pi(x_t \mid s_t; \theta) \Psi_t) \nabla_\theta \log q_i(\tau; \theta)^\mathsf{T} + \sum_{t=0}^{H} \nabla_\theta^2 \log \pi(x_t \mid s_t; \theta) \Psi_t \right\|
$$
$$
\leq \frac{(H+1)RG^2 + RL}{(1-\gamma)^2}.
$$

**Proof of (iii):** Note that

$$
\| \nabla_\theta^2 J_i(\theta_1) - \nabla_\theta^2 J_i(\theta_2) \|
$$
$$
\leq \left\| \sum_{t=0}^{H} \nabla_\theta \log \pi(x_t \mid s_t; \theta_1) \Psi_t \nabla_\theta \log q_i(\tau; \theta_1)^\mathsf{T} - \sum_{t=0}^{H} \nabla_\theta \log \pi(x_t \mid s_t; \theta_2) \Psi_t \nabla_\theta \log q_i(\tau; \theta_2)^\mathsf{T} \right\|
$$
$$
+ \left\| \sum_{t=0}^{H} \nabla_\theta^2 \log \pi(x_t \mid s_t; \theta_1) \Psi_t - \sum_{t=0}^{H} \nabla_\theta^2 \log \pi(x_t \mid s_t; \theta_2) \Psi_t \right\|
$$
$$
\leq \| \nabla_\theta \log q_i(\tau; \theta) \| \left\| \sum_{t=0}^{H} \nabla_\theta \log \pi(x_t \mid s_t; \theta_1) \Psi_t - \sum_{t=0}^{H} \nabla_\theta \log \pi(x_t \mid s_t; \theta_2) \Psi_t \right\|
$$
$$
+ \left\| \sum_{t=0}^{H} \nabla_\theta \log \pi(x_t \mid s_t; \theta_1) \Psi_t \right\| \left\| \nabla_\theta \log q_i(\tau; \theta_1) - \nabla_\theta \log q_i(\tau; \theta_2) \right\|
$$
$$
+ \left\| \sum_{t=0}^{H} \nabla_\theta^2 \log \pi(x_t \mid s_t; \theta_1) \Psi_t - \sum_{t=0}^{H} \nabla_\theta^2 \log \pi(x_t \mid s_t; \theta_2) \Psi_t \right\|.
$$

First, we consider the Lipschitz parameter of $\sum_{t=0}^{H} \nabla_\theta \log \pi(x_t \mid s_t; \theta) \Psi_t$:

$$
\left\| \sum_{t=0}^{H} \nabla_\theta \log \pi(x_t \mid s_t; \theta_1) \Psi_t - \sum_{t=0}^{H} \nabla_\theta \log \pi(x_t \mid s_t; \theta_2) \Psi_t \right\|
$$
$$
\leq \sum_{t=0}^{H} \| \nabla_\theta \log \pi(x_t \mid s_t; \theta_1) - \nabla_\theta \log \pi(x_t \mid s_t; \theta_2) \| |\Psi_t|.
$$

According to Assumption 1, the gradient of logarithmic policy is smooth with parameter $L$, i.e.,

$$
\| \nabla_\theta \log \pi(x_t \mid s_t; \theta_1) - \nabla_\theta \log \pi(x_t \mid s_t; \theta_2) \| \leq L \| \theta_1 - \theta_2 \|.
$$

Therefore,

$$
\left\| \sum_{t=0}^{H} \nabla_\theta \log \pi(x_t \mid s_t; \theta_1) \Psi_t - \sum_{t=0}^{H} \nabla_\theta \log \pi(x_t \mid s_t; \theta_2) \Psi_t \right\|
$$
$$
\leq L \| \theta_1 - \theta_2 \| \sum_{t=0}^{H} \frac{R\gamma^{t'}}{1-\gamma}
$$
$$
\leq \frac{RL}{(1-\gamma)^2} \| \theta_1 - \theta_2 \|.
$$

It is obvious that $\nabla_\theta \log q_i(\tau; \theta)$ is Lipschitz with parameter $(H+1)L$, i.e.,

$$
\| \nabla_\theta \log q_i(\tau; \theta_1) - \nabla_\theta \log q_i(\tau; \theta_2) \| \leq (H+1)L \| \theta_1 - \theta_2 \|.
$$

According to Assumption 2, wherein the gradient of the logarithmic policy is smooth with parameter $K$, we have a similar conclusion as in the above proof:

$$
\left\| \sum_{t=0}^{H} \nabla_\theta^2 \log \pi(x_t \mid s_t; \theta_1) \Psi_t - \sum_{t=0}^{H} \nabla_\theta^2 \log \pi(x_t \mid s_t; \theta_2) \Psi_t \right\| \leq \frac{RK}{(1-\gamma)^2} \| \theta_1 - \theta_2 \|.
$$

From the proof of Lemma 1(ii), we know the bound $\|\nabla_\theta \log q_i(\tau; \theta)\| \le (H+1)G$. The result in Lemma 1(i) shows that $\|\sum_{t=0}^{H} \nabla_\theta \log \pi(x_t \mid s_t; \theta) \Psi_t\| \le \frac{RG}{(1-\gamma)^2}$. Finally, these yield the result that

$$\|\nabla_\theta^2 J_i(\theta_1) - \nabla_\theta^2 J_i(\theta_2)\| \le \left( (H+1)G \frac{RL}{(1-\gamma)^2} + \frac{RG}{(1-\gamma)^2}(H+1)L + \frac{RK}{(1-\gamma)^2} \right) \|\theta_1 - \theta_2\|$$
$$= \frac{2(H+1)RGL + RK}{(1-\gamma)^2} \|\theta_1 - \theta_2\|.$$

### E.3. Proof of Lemma 2

In this section, we show the proof details of Lemma 2.

**Proof of (i):** We first note that

$$\|\nabla_\theta F(\theta)\| = \|(I + \alpha \widehat{\nabla}_\theta^2 J_i(\theta, \widehat{\mathcal{D}}_{\exp}^i)) \nabla_\theta J_i(\theta + \alpha \widehat{\nabla} J_i(\theta, \widehat{\mathcal{D}}_{\exp}^i))$$
$$+ J_i(\theta + \alpha \widehat{\nabla} J_i(\theta, \widehat{\mathcal{D}}_{\exp}^i)) \sum_{\tau \in \widehat{\mathcal{D}}_{\exp}^i} \sum_{t=0}^{H} \nabla_\theta \log \pi_\theta(x_t \mid s_t; \theta)\|$$
$$\le \|I + \alpha \widehat{\nabla}_\theta^2 J_i(\theta, \widehat{\mathcal{D}}_{\exp}^i)\| \|\nabla_\theta J_i(\theta + \alpha \widehat{\nabla} J_i(\theta, \widehat{\mathcal{D}}_{\exp}^i))\|$$
$$+ \|J_i(\theta + \alpha \widehat{\nabla} J_i(\theta, \widehat{\mathcal{D}}_{\exp}^i))\| \left\| \sum_{\tau \in \widehat{\mathcal{D}}_{\exp}^i} \sum_{t=0}^{H} \nabla_\theta \log \pi_\theta(x_t \mid s_t; \theta) \right\|.$$

Lemma 1 implies that $\|\nabla_\theta J_i(\theta - \alpha \widehat{\nabla} J_i(\theta, \widehat{\mathcal{D}}_{\exp}^i))\| \le \eta_G$. For any parameter $\theta$, the accumulated reward function is bounded by

$$\|J_i(\theta)\| = \|\sum_{t=0}^{H} \gamma^t R_i(s_t, x_t)]\|$$
$$\le R \sum_{t=0}^{H} \gamma^t$$
$$\le \frac{R}{1-\gamma}.$$

Recalling Assumption 1, we have that $\|\sum_{\tau \in \widehat{\mathcal{D}}_{\exp}^i} \sum_{t=0}^{H} \nabla_\theta \log \pi_\theta(s_t, x_t; \theta)\|$ is bounded by $GD(H+1)$. $(I + \alpha \widehat{\nabla}_\theta^2 J_i(\theta, \widehat{\mathcal{D}}_{\exp}^i))$ is bounded by $1 + \alpha \eta_H$. Relying on the assumption $\alpha \le \eta_H$, we know $(1 + \alpha \eta_H) \le 2$. Now, we know that the gradient of the objective function $\|\nabla_\theta F(\theta)\|$ is bounded by $2\eta_G + \frac{(H+1)DRG}{1-\gamma} = \frac{2RG}{(1-\gamma)^2} + \frac{D(H+1)RG}{1-\gamma}$.

**Proof of (ii):**

The Lipschitz parameter of $\widehat{\nabla}_\theta F(\theta)$ is the sum of the Lipschitz parameters of $(I + \alpha \widehat{\nabla}_\theta^2 J_i(\theta, \widehat{\mathcal{D}}_{\exp}^i)) \nabla_\theta J_i(\theta + \alpha \widehat{\nabla} J_i(\theta, \widehat{\mathcal{D}}_{\exp}^i))$ and $J_i(\theta + \alpha \widehat{\nabla} J_i(\theta, \widehat{\mathcal{D}}_{\exp}^i)) \sum_{\tau \in \widehat{\mathcal{D}}_{\exp}^i} \sum_{t=0}^{H} \nabla_\theta \log \pi(x_t \mid s_t; \theta)$. Next, we analyze each item separately.

Consider the Lipschitz parameter of $(I + \alpha \widehat{\nabla}_\theta^2 J_i(\theta, \widehat{\mathcal{D}}_{\exp}^i)) \nabla_\theta J_i(\theta + \alpha \widehat{\nabla} J_i(\theta, \widehat{\mathcal{D}}_{\exp}^i))$. We have

$$\|(I + \alpha \widehat{\nabla}_\theta^2 J_i(\theta_1, \widehat{\mathcal{D}}_{\exp}^i)) \nabla_\theta J_i(\theta_1 + \alpha \widehat{\nabla} J_i(\theta_1, \widehat{\mathcal{D}}_{\exp}^i))$$
$$- (I + \alpha \widehat{\nabla}_\theta^2 J_i(\theta_2, \widehat{\mathcal{D}}_{\exp}^i)) \nabla_\theta J_i(\theta_2 + \alpha \widehat{\nabla} J_i(\theta_2, \widehat{\mathcal{D}}_{\exp}^i))\|$$
$$\le \|(I + \alpha \widehat{\nabla}_\theta^2 J_i(\theta, \widehat{\mathcal{D}}_{\exp}^i))\| \|\nabla_\theta J_i(\theta_1 + \alpha \widehat{\nabla} J_i(\theta_1, \widehat{\mathcal{D}}_{\exp}^i)) - \nabla_\theta J_i(\theta_2 + \alpha \widehat{\nabla} J_i(\theta_2, \widehat{\mathcal{D}}_{\exp}^i))\|$$
$$+ \|\nabla_\theta J_i(\theta + \alpha \widehat{\nabla} J_i(\theta, \widehat{\mathcal{D}}_{\exp}^i))\| \|\alpha \widehat{\nabla}_\theta^2 J_i(\theta_1, \widehat{\mathcal{D}}_{\exp}^i) - \alpha \widehat{\nabla}_\theta^2 J_i(\theta_2, \widehat{\mathcal{D}}_{\exp}^i)\|.$$

According to the result in Lemma 1, we know that $(I + \alpha \widehat{\nabla}_\theta^2 J_i(\theta, \widehat{\mathcal{D}}_{\exp}^i))$ is bounded by $(1 + \alpha \eta_H)$ and smooth with parameter $\alpha \eta_\rho$. $\nabla_\theta J_i(\theta)$ is bounded by $\eta_G$ and smooth with parameter $\eta_H$. Along with the fact that the Lipschitz parameter

of the combination of functions is the product of their Lipschitz parameters and $\theta + \alpha \widehat{\nabla} J_i(\theta, \widehat{\mathcal{D}}_{\exp}^i)$ is smooth with parameter $1 + \alpha \eta_H$, $\nabla_\theta J_i(\theta + \alpha \widehat{\nabla} J_i(\theta, \widehat{\mathcal{D}}_{\exp}^i))$ is smooth with parameter $(1 + \alpha \eta_H) \eta_H$. Therefore,

$$
\begin{aligned}
&\|(I + \alpha \widehat{\nabla}_\theta^2 J_i(\theta_1, \widehat{\mathcal{D}}_{\exp}^i)) \nabla_\theta J_i(\theta_1 + \alpha \widehat{\nabla} J_i(\theta_1, \widehat{\mathcal{D}}_{\exp}^i)) \\
&- (I + \alpha \widehat{\nabla}_\theta^2 J_i(\theta_2, \widehat{\mathcal{D}}_{\exp}^i)) \nabla_\theta J_i(\theta_2 + \alpha \widehat{\nabla} J_i(\theta_2, \widehat{\mathcal{D}}_{\exp}^i))\| \\
&\leq (1 + \alpha \eta_H)(1 + \alpha \eta_H) \eta_H \|\theta_1 - \theta_2\| + \eta_G(\alpha \eta_\rho) \|\theta_1 - \theta_2\| \\
&= ((1 + \alpha \eta_H)^2 \eta_H + \alpha \eta_G \eta_\rho) \|\theta_1 - \theta_2\|.
\end{aligned}
$$

Using the assumption $\alpha \leq \eta_H$, we know $(1 + \alpha \eta_H) \leq 2$. Consequently, $(I + \alpha \widehat{\nabla}_\theta^2 J_i(\theta, \widehat{\mathcal{D}}_{\exp}^i)) \nabla_\theta J_i(\theta + \alpha \widehat{\nabla}_\theta J_i(\theta, \widehat{\mathcal{D}}_{\exp}^i))$ is smooth with parameter $4\eta_H + \alpha \eta_G \eta_\rho$.

Now consider the Lipschitz parameter of $J_i(\theta + \alpha \widehat{\nabla} J_i(\theta, \widehat{\mathcal{D}}_{\exp}^i)) \sum_{\tau \in \widehat{\mathcal{D}}_{\exp}^i} \sum_{t=0}^{H} \nabla_\theta \log \pi(x_t \mid s_t; \theta)$:

$$
\begin{aligned}
&\|J_i(\theta_1 + \alpha \widehat{\nabla} J_i(\theta_1, \widehat{\mathcal{D}}_{\exp}^i)) \sum_{\tau \in \widehat{\mathcal{D}}_{\exp}^i} \sum_{t=0}^{H} \nabla_\theta \log \pi(x_t \mid s_t; \theta_1) \\
&- J_i(\theta_2 + \alpha \widehat{\nabla} J_i(\theta_2, \widehat{\mathcal{D}}_{\exp}^i)) \sum_{\tau \in \widehat{\mathcal{D}}_{\exp}^i} \sum_{t=0}^{H} \nabla_\theta \log \pi(x_t \mid s_t; \theta_2)\| \\
&\leq \|J_i(\theta + \alpha \widehat{\nabla} J_i(\theta, \widehat{\mathcal{D}}_{\exp}^i))\| \sum_{\tau \in \widehat{\mathcal{D}}_{\exp}^i} \sum_{t=0}^{H} \|\nabla_\theta \log \pi(x_t \mid s_t; \theta_1) - \nabla_\theta \log \pi(x_t \mid s_t; \theta_2)\| \\
&+ \sum_{\tau \in \widehat{\mathcal{D}}_{\exp}^i} \sum_{t=0}^{H} \|\nabla_\theta \log \pi(x_t \mid s_t; \theta)\| \|J_i(\theta_1 + \alpha \widehat{\nabla} J_i(\theta_1, \widehat{\mathcal{D}}_{\exp}^i)) - J_i(\theta_2 + \alpha \widehat{\nabla} J_i(\theta_2, \widehat{\mathcal{D}}_{\exp}^i))\|.
\end{aligned}
$$

Relying on the Assumption 2, we know that $\nabla_\theta \log \pi(x_t \mid s_t; \theta)$ is bounded by $G$ and smooth with parameter $L$. Along with the fact that the Lipschitz parameter of the combination of functions is the product of their Lipschitz parameters and $\theta + \alpha \widehat{\nabla} J_i(\theta, \widehat{\mathcal{D}}_{\exp}^i)$ is smooth with parameter $1 + \alpha \eta_H$, $J_i(\theta + \alpha \widehat{\nabla} J_i(\theta, \widehat{\mathcal{D}}_{\exp}^i))$ is smooth with parameter $(1 + \alpha \eta_H) \eta_G \leq 2\eta_G$. Therefore,

$$
\begin{aligned}
&\|J_i(\theta_1 + \alpha \widehat{\nabla} J_i(\theta_1, \widehat{\mathcal{D}}_{\exp}^i)) \sum_{\tau \in \widehat{\mathcal{D}}_{\exp}^i} \sum_{t=0}^{H} \nabla_\theta \log \pi(x_t \mid s_t; \theta_1) \\
&- J_i(\theta_2 + \alpha \widehat{\nabla} J_i(\theta_2, \widehat{\mathcal{D}}_{\exp}^i)) \sum_{\tau \in \widehat{\mathcal{D}}_{\exp}^i} \sum_{t=0}^{H} \nabla_\theta \log \pi(x_t \mid s_t; \theta_2)\| \\
&\leq \frac{R}{1 - \gamma} D(H + 1) L \|\theta_1 - \theta_2\| + D(H + 1) G 2\eta_G \|\theta_1 - \theta_2\| \\
&= D(H + 1) R \left( \frac{L}{1 - \gamma} + \frac{2G^2}{(1 - \gamma^2)} \right).
\end{aligned}
$$

According to the following derivation, we know that the Lipschitz parameter of $J_i(\theta + \alpha \widehat{\nabla} J_i(\theta, \widehat{\mathcal{D}}_{\exp}^i)) \sum_{\tau \in \widehat{\mathcal{D}}_{\exp}^i} \sum_{t=0}^{H} \nabla_\theta \log \pi(x_t \mid s_t; \theta)$ is $D(H + 1) R \left( \frac{L}{1-\gamma} + \frac{2G^2}{(1-\gamma^2)} \right)$.

Finally, the Lipschitz parameter of $\nabla_\theta F(\theta)$ is $4\eta_H + \alpha \eta_G \eta_\rho + D(H + 1) R(\frac{L}{1-\gamma} + \frac{2G^2}{(1-\gamma^2)})$.

### E.4. Proof of Theorem 1

First, we establish an upper bound on the variance of the estimation of the meta-objective function gradient $\nabla_\theta F(\theta)$.

**Lemma 3.** *Suppose that the conditions in Assumptions.1, 2 are satisfied. For the case that $\alpha \in (0, \frac{1}{\eta_H}]$, and any choice of task subset $\mathcal{B}$, we have*

$$\mathbb{E}\|\widehat{\nabla}_\theta F(\theta) - \nabla_\theta F(\theta)\| \leq \frac{L_G^2}{BD}.$$

The proof is based on an application of the law of large numbers and variance additivity. If $\{X_1, X_2, \ldots, X_n\}$ are independent random variables with $\mathbb{E}[X_i] = \mu$, and variance bounded by $\text{Var}[X_i] \leq \sigma^2$, then the variance of the sample mean is bounded by

$$\mathbb{E}\left[\left\|\frac{X_1 + \cdots + X_n}{n} - \mu\right\| \leq \frac{\sigma^2}{n}\right].$$

Next, we proceed with the proof. Using the smoothness property of $\nabla_\theta F(\theta)$, we have

$$|F(\theta_{k+1}) - F(\theta_k) - \nabla_\theta F(\theta_k) \times (\theta_{k+1} - \theta_k)| \leq \frac{L_H^2}{2}\|\theta_{k+1} - \theta_k\|.$$

At iteration $k+1$, we have $\theta_{k+1} - \theta_k = \beta\widehat{\nabla}_\theta F(\theta_k)$, and therefore,

$$-F(\theta_{k+1}) \leq -F(\theta_k) - \beta\nabla_\theta F(\theta_k) \times \widehat{\nabla}_\theta F(\theta_k) + \frac{L_H^2}{2}\beta^2\|\widehat{\nabla}_\theta F(\theta_k)\|^2.$$

Taking the expectations of both sides, we obtain

$$-\mathbb{E}[F(\theta_{k+1})] \leq -\mathbb{E}[F(\theta_k)] - \beta\mathbb{E}[\|\nabla_\theta F(\theta_k)\|^2]$$
$$+ \frac{L_H^2}{2}\beta^2(\mathbb{E}[\|\nabla_\theta F(\theta_k)\|^2] + \mathbb{E}[\|\widehat{\nabla}_\theta F(\theta) - \nabla_\theta F(\theta)\|^2])$$
$$\leq -\mathbb{E}[F(\theta_k)] - \frac{\beta}{2}\mathbb{E}[\|\nabla_\theta F(\theta_k)\|^2] + \frac{L_G^2 L_H \beta^2}{2BD}.$$

We prove the conclusion by contradiction. Assume our result does not hold for the first $T$ iterations, i.e.,

$$\mathbb{E}[\|\nabla_\theta F(\theta_k)\|^2] \geq \frac{2L_G^2 L_H \beta}{BD} + \epsilon^2.$$

For any $0 \leq k \leq T - 1$, we have

$$-\mathbb{E}[F(\theta_{k+1})] \leq -\mathbb{E}[F(\theta_k)] - \frac{\beta\epsilon^2}{2} - \frac{L_G^2 L_H \beta^2}{BD} + \frac{L_G^2 L_H \beta^2}{2BD}.$$

Summing up the above formulation for $k = 0, \ldots, T - 1$, we obtain

$$-\mathbb{E}[F(\theta_T)] \leq -\mathbb{E}[F(\theta_0)] - T(\frac{\beta\epsilon^2}{2} + \frac{L_G^2 L_H \beta^2}{2BD}).$$

We know that $\mathbb{E}[F(\theta)] \in [\frac{a}{1-\gamma}, \frac{b}{1-\gamma}]$, and hence $\mathbb{E}[F(\theta_0)] - \mathbb{E}[F(\theta_T)] \leq \frac{b-a}{1-\gamma}$. Then, we have

$$T\left(\frac{\beta\epsilon^2}{2} + \frac{L_G^2 L_H \beta^2}{2BD}\right) \leq \frac{b-a}{1-\gamma}.$$

When we choose $T \geq \frac{b-a}{1-\gamma}(\frac{2}{\beta\epsilon^2} + \frac{2BD}{L_G^2 L_H \beta^2})$, contradiction occurs. Hence, the desired result follows.

