# OpenReview forum: "Counterfactual Bootstrap for Robust Meta-Reinforcement Learning"
_ICML.cc/2026/Conference — ICML 2026 regular_

### Official Review · Reviewer_M8Vp · 2026-03-11

**Soundness:** 2
**Presentation:** 3
**Significance:** 2
**Originality:** 3
**Overall Recommendation:** 4
**Confidence:** 3

**Summary:**

The paper studies Meta-Reinforcement Learning (meta-RL) under unobserved confounding, relaxing the common assumption that actions, states, and rewards are not jointly influenced by hidden variables. To relax this assumption, the authors introduce a novel robust augmentation procedure that leverages confounded observational data to predict non-identifiable system dynamics of the source domains while generating new counterfactual trajectories for training a meta-policy with enhanced adaptability across confounded environments. The proposed augmentation framework is instantiated within MAML in the main paper and applied to other meta-RL methods in the Appendix.

**Compliance With Llm Reviewing Policy:**

Affirmed.

**Final Justification:**

I maintain my original score.

**Key Questions For Authors:**

1.The proposed framework critically depends on the posterior distribution used to generate counterfactual trajectories. How robust is the method to posterior misspecification? For example, if the posterior is constructed using incorrect assumptions or noisy information about the environment, how significantly would this affect the resulting policy initialization?

2.The experimental evaluation appears relatively limited. Could the authors provide additional experiments under more diverse settings (e.g., varying degrees or types of unobserved confounding) to better understand how the method behaves when the strength of confounding changes? In particular, have the authors considered settings where the unobserved confounding actually favors conventional methods? In such scenarios, does the proposed approach still maintain an advantage, or could its performance deteriorate relative to standard baselines? Empirical results or discussion on this case would help clarify the robustness and practical applicability of the method.

3.The method seems to implicitly assume a stable confounding structure. How robust is the approach when the unobserved confounders change over time or across environments?

4.The related work section does not sufficiently discuss prior literature that leverages posterior distributions to improve learning performance.

**Limitations:**

The paper would benefit from a clearer discussion of the limitations of the posterior assumptions and potential risks when the confounding structure or environment is misspecified.

**Strengths And Weaknesses:**

Soundness. The paper appears technically reasonable overall. However, beyond the proposed approach, alternative ways of specifying the posterior distribution could be considered, and the potential implications of these alternatives should be discussed. In addition, for experiments, more comprehensive experimental settings would be helpful to better assess the impact of unobserved confounders.

Presentation. The paper is generally well structured and clearly organized. However, Figure 4 would benefit from additional explanation. For the discussion of prior and concurrent work, related literature that leverages posterior distributions to improve method performance is not discussed in sufficient depth.

Significance. The paper demonstrates the proposed counterfactual data augmentation framework primarily within MAML, with extensions to other meta-RL methods discussed in the appendix. However, the broader implications of unobserved confounders (positive and negative) and their influence on learning performance are not explored in sufficient depth.

Originality. The work offers an interesting perspective for reinforcement learning by explicitly considering the impact of unobserved confounders. However, the discussion of closely related literature is needed. The broader implications of unobserved confounders (positive and negative) and their influence on learning performance are not explored in sufficient depth.

---

> ### Author Rebuttal · Authors · 2026-03-31
>
> We thank you for your thoughtful feedback. We appreciate your acknowledgment of the presentation and novelty of our work and address your concerns in the sequel.
>
> > **Soundness: Alternative posterior specifications and more comprehensive experiments.**
>
> We do not sample from a posterior $\rho(\mathcal{M}|\mathcal{D}^i_{obs})$ directly. Instead, we derive non-parametric Manski bounds on $\mathcal{T}_i$ (Eqs. 5–6) and $\mathcal{R}_i$ (Eqs. 7–8), then uniformly sample candidate models within these bounds — making no assumptions about the prior $\rho$. Alternative strategies (e.g., pessimistic or optimistic sampling within non-parametric bounds) are possible; we will discuss these trade-offs in the revision. Experimental breadth is addressed under Q2.
>
>
> > **Presentation: Figure 4 and related work.**
>
> We will expand Figure 4's caption: "The optimal initialization $\theta$ corresponds to the true tasks $\mathcal{M}_1, \mathcal{M}_2, \mathcal{M}_3$. Naïve meta-RL converges to $\tilde{\theta}$ corresponding to biased tasks $\tilde{\mathcal{M}}_i$. Dashed regions represent equivalence classes of models compatible with each task's data. Our causal approach samples from these classes, yielding $\hat{\theta}$ closer to the optimum."
>
> We will add a dedicated related work section discussing posterior-based methods in RL and meta-learning (see Q4).
>
>
>
> >**Significance / Originality: Broader implications of confounders.**
>
> We appreciate this point. Unobserved confounders can have asymmetric effects: (i) *negative confounding* introduces spurious correlations that mislead policy learning (as in our Windy Gridworld, where the demonstrator's wind-awareness creates illusory shortcuts), while (ii) *benign confounding* may coincidentally align observational and interventional distributions, in which case standard methods perform adequately. Our framework handles both cases: when confounding is strong, the Manski bounds are wide and counterfactual sampling corrects for bias; when confounding is known to be weak, the bounds will be tightened around the observational estimates and our method recovers standard meta-RL behavior. We will add this discussion to the revised manuscript.
>
>
>
>
> > **Q1: Robustness to posterior misspecification.**
>
> Since we derive **non-parametric bounds** directly from observational data (Eqs. 5–8) rather than specifying a parametric posterior, our sampling approach is immune to posterior misspecification in the traditional sense. The bounds are guaranteed to contain the true CMDP model, regardless of the model prior. Simulations suggest this non-parametric approximation performs well in practice.
>
>
>
> > **Q2: Varying confounding strength; settings favoring conventional methods.**
>
> The degree of confounding in our environments is controlled by the wind distribution, which varies based on the grid position. In our current setup, the probability of no-wind in lava passages is 0.1 (strong confounding). We will add experiments with weak confounding to show the full spectrum. Under weak confounding, the observational and interventional distributions nearly coincide, so  conventional meta-RL methods and our causally enhanced methods exhibit comparable performance. Note that our experiments also include instances where confounding favors conventional methods, e.g., the purple key task in Fig. 5 (a). The simulation results report the cumulative reward averaging over target tasks, i.e., there are more tasks compatible with observations where the confounding bias is harmful for learning.
>
>
>
>
> > **Q3: Time-varying or environment-varying confounding structure.**
>
> Our bounds (Eqs. 5–8) are computed per-task from each task's observational data, so they naturally accommodate heterogeneous confounding across environments. If the confounding structure differs between tasks $\mathcal{M}_i$ and $\mathcal{M}_j$, each task's bounds reflect its own confounding regime. For time-varying confounding within a single episode,  our framework approximates this setting by (1) creating separate exogenous variables for each state; and (2) letting state values pick the active confounding variable. For example, in the Windy Minigrid (Fig. 3), the distribution over wind directions is state-dependent; the wind is more likely to blow for the shorter route (risky) than the longer route (safe).
>
>
>
> > **Q4: Related work on posterior-based learning.**
>
> We will add a section covering: (i) posterior sampling for RL exploration (Osband et al., 2013), which maintains MDP beliefs but assumes no confounding; (ii) Bayesian meta-learning (Yoon et al., 2018; Grant et al., 2018); and (iii) task inference methods (Rakelly et al., 2019; Zintgraf et al., 2020). Our approach differs fundamentally: rather than a probabilistic belief that could be misspecified, we construct a guaranteed feasible region via partial identification, requiring no parametric assumptions on the prior or confounding mechanism.

---

> > ### Author Rebuttal · Reviewer_M8Vp · 2026-04-03
> >
> > I acknowledge the authors’ rebuttal. However, my concern remains only partially addressed. In particular, the experimental evaluation remains limited, and the relevant results have not been updated.

---

> > > ### Author Response · Authors · 2026-04-03
> > >
> > > Thank you for the continued engagement. We have now completed the additional experiment you requested.
> > >
> > > > "My concern remains only partially addressed. In particular, the experimental evaluation remains limited, and the relevant results have not been updated."
> > >
> > > We ran ablations in a weak/benign confounding environment by setting the no-wind probability to 0.9 in the lava passages. The results on the Pick-Up-Key task are:
> > >
> > > | Confounding | No-wind prob. | Causal-MAML | MAML | PPO |
> > > |-------------|--------------|-------------|------|-----|
> > > | Strong | 0.10 | **1.21** |0.05 | 0.65 |
> > > | Weak | 0.90 | **0.82** | 0.80 | 0.66 |
> > >
> > > These results confirm three important properties of our method:
> > >
> > > (1) Robustness under strong confounding. When confounding is severe (no-wind = 0.1), standard MAML fails entirely (negative reward), while Causal-MAML substantially outperforms even vanilla PPO. This is the core setting motivating our work.
> > >
> > > (2) Graceful degradation. As confounding weakens, the causal bounds tighten. Correspondingly, the performance gap between Causal-MAML and standard MAML narrows, as the theory predicts.
> > >
> > > (3) No performance penalty in benign settings. Under weak confounding (no-wind = 0.9), Causal-MAML performs comparably to MAML. This confirms that our augmentation does not hurt when confounding is negligible — the method gracefully recovers standard meta-RL behavior rather than introducing unnecessary conservatism.
> > >
> > > We will include the full ablation table and discussion in the revised manuscript.

---

### Official Review · Reviewer_W5Dk · 2026-03-11

**Soundness:** 2
**Presentation:** 3
**Significance:** 3
**Originality:** 3
**Overall Recommendation:** 4
**Confidence:** 3

**Summary:**

The authors proposed an augmentation method for meta-reinforcement learning with confounded observational data.

**Compliance With Llm Reviewing Policy:**

Affirmed.

**Final Justification:**

My questions are addressed, and I maintain my positive score.

**Key Questions For Authors:**

My main concern is the lack of theoretical guarantees. Beyond the local-optimum solution, a more important question is why the proposed solution is expected to perform better than the naive solution, beyond the examples and heuristics presented in the paper. For instance, sampling from $\rho(\mathcal{M}|\mathcal{D}^i_{\mathrm{obs}})$ is related to counterfactual generation, which may help mitigate distribution shift caused by unobserved confounding. A more formal justification of the conditions under which the proposed augmentation is expected to work can be helpful.

**Limitations:**

Some limitations are discussed.

**Strengths And Weaknesses:**

The problem is well-motivated and important for both causal inference and reinforcement learning. The presentation is clear.

---

> ### Author Rebuttal · Authors · 2026-03-31
>
> We are grateful for your comments and for acknowledging the presentation, importance, and novelty of our work. We have addressed your question as follows.
>
> > **Q1: Why is the proposed solution expected to outperform the naïve approach? Formal justification needed.**
>
> This is an excellent question. We provide a formal argument in four steps.
>
> **The naïve approach is provably biased.** When unobserved confounders exist, the observational distribution $P(s'|s,x)$ conflates the causal effect of action $x$ with the confounder's influence. The stochastic gradient $\tilde{\nabla}J_i(\theta, \mathcal{D}^i_{obs})$ computed from confounded data is therefore a biased estimate of the true policy gradient $\nabla J_i(\theta)$. Optimizing Eq. (3) converges to a biased solution $\tilde{\theta}$ that may perform arbitrarily worse than the optimal $\theta$ — as demonstrated empirically in Figure 1b, where MAML, Pretrained-PPO, and RL² all fail to outperform vanilla PPO.
>
> **Our feasible region is guaranteed to contain the true model.** The Manski bounds (Eqs. 5–8) are derived without parametric assumptions on the confounder distribution $P(U)$. For any CMDP $\mathcal{M}\_i$ generating the observational data, its true transition $\mathcal{T}_i$ and reward $\mathcal{R}_i$ must lie within these bounds. This is not a heuristic — it follows directly from the principle of partial causal identification (see our derivation in the W1 response to Reviewer 1). Corollary 1 further ensures that restricting to canonical CMDPs incurs no loss of generality.
>
> **Averaging over the feasible region yields an unbiased meta-objective.** By sampling candidate models $\hat{\mathcal{M}}\_i$ uniformly from the feasible region and collecting experimental trajectories $\hat{\mathcal{D}}^i_{exp}$ via interaction with these models, the stochastic gradient $\hat{\nabla}J_i(\theta, \hat{\mathcal{D}}^i_{exp})$ is an unbiased estimate of the gradient of the augmented objective $\hat{F}(\theta)$ (Eq. 4). Since the true model is contained in the feasible region, the augmented objective $\hat{F}(\theta)$ includes the true task in expectation. Theorem 1 then guarantees convergence of this unbiased procedure to a first-order stationary point, with sample complexity $O(\epsilon^{-4})$.
>
> **Conditions for the advantage to hold.** The improvement over naïve meta-RL is most pronounced when: (i) the behavioral policy has broad support, so bounds are tight for most state-action pairs; and (ii) confounding is sufficient enough that the naïve gradient is substantially biased. We will formalize this comparison and add it to the revised manuscript.

---

> > ### Author Rebuttal · Reviewer_W5Dk · 2026-04-02
> >
> > Thank you for the response. My concerns are addressed, and I will maintain my current score.

---

> > > ### Author Response · Authors · 2026-04-02
> > >
> > > We are pleased to hear that our rebuttal has addressed your concerns, and we appreciate your positive feedback. If you have any further questions, please don’t hesitate to comment, and we will be happy to assist.

---

### Official Review · Reviewer_fRNP · 2026-03-16

**Soundness:** 3
**Presentation:** 3
**Significance:** 2
**Originality:** 3
**Overall Recommendation:** 4
**Confidence:** 3

**Summary:**

For a sequential decision-making task, a learner might have access to a set of offline observational datasets from similar tasks. Such data can be leveraged to solve the task at hand more efficiently, through pre-training or meta-learning (Meta-RL). The paper identifies a major issue when naively applying such Meta-RL approaches in settings where unobserved confounders are present in the offline data, i.e., when the offline demonstrators can observe more than what the learner can see. In other words, the relevant transition/reward structure of the tasks cannot be identified from offline observational data alone. They propose a remedy by changing the typical model-agnostic meta-learning (MAML) loss to one that takes the non-identifiability into account. In particular, for each task in the offline data, they use partial identification (Manski bounds) to derive the set of plausible dynamics that are consistent with the observational data for that task, and then minimize the MAML loss, taking expectation over all the consistent models. The final outcome is a policy network with an initialization that leads to a more efficient adaptation. They show better performance (achieving higher expected rewards) in three variants of the Windy Gridworld environment, compared to vanilla meta-RL approaches, or standard PPO algorithm that ignores the offline data.

**Compliance With Llm Reviewing Policy:**

Affirmed.

**Final Justification:**

I thank the authors for providing detailed responses to my questions/comments. I increase the presentation score conditioned on updating the manuscript based on the discussion we had. However, I'll keep my overall recommendation as it is, mostly due to limited significance of the theoretical results. Still, I recommend for the acceptance of the paper.

**Key Questions For Authors:**

Please refer to the questions asked under Significance, Originality, and Soundness above.

**Limitations:**

The paper discusses limitations under Impact Statement.

**Strengths And Weaknesses:**

## Significance, Originality, and Soundness

I have not checked all the details in the proofs and the derivation of the gradients and hyper-gradients. However, I can tell the idea itself makes sense, and as far as I know, it is novel in this context. In summary, the idea is to imagine all the consistent MDPs with the offline data during the offline training step, and come up with an initialization of the policy network that once a step of policy gradient is applied, the expected return over all such consistent MDPs will be increased. However, I do have a few concerns that I hope the authors can address:

1. Equations (5) to (8) are stated without a proof. The authors only refer to (Manski,1990), but the results there are not specifically proved with the notation in the paper. E.g., based on equations (7) and (8), if $p(x|s) = 1$, i.e., if the demonstrator always takes action $x$ at state $s$, the reward $\mathcal{R}_i(s, x)$ will be identified and equal to $\mathbb{E}[Y_t | S_t = s, X_t = x]$. The paper should explain more clearly why the bounds collapse under deterministic logging and what exact causal assumptions make that collapse valid.

2. Theorem 1 only shows convergence to a local optimum of the objective function $F(\theta)$. In the abstract, the authors mention that _"Theoretical analysis reveals that our causal Meta-RL approach is guaranteed to yield a solution that minimizes generalization loss in future inference tasks"_. These two statements are not the same. The theory does not say anything about the nature of the learned initialization for a future task. For example, it's likely that the partial ID bounds would be vacuous and will include MDPs with opposite transition/reward functions. In such scenarios, the expected gradients will cancel out, resulting in an uninformative initialization. The current theory fails to capture such nuances. Could the authors provide more insights on such cases?

3. The paper lacks a dedicated related work sections. There current discussion in the introduction omits several papers that address similar questions (although the settings might not be exactly the same). I think the authors can frame their contribution better if compared to those. I include a few below:

[AA] Bellot, Alexis, Alan Malek, and Silvia Chiappa. "Transportability for bandits with data from different environments." Advances in Neural Information Processing Systems 36 (2023): 44356-44381.

[BB] Li, Qiyang, et al. "Accelerating exploration with unlabeled prior data." Advances in Neural Information Processing Systems 36 (2023): 67434-67458.

[CC] Balazadeh, Vahid, et al. "Sequential decision making with expert demonstrations under unobserved heterogeneity." Advances in Neural Information Processing Systems 37 (2024): 65476-65498.

**A Suggestion:**

As the authors already discussed and showed in Appendix A, the idea itself is not inherently related to MAML and can be used with other approaches. They included methods like pretrained PPO and $\text{RL}^2$. However, this still does not touch on the more modern approaches that use transformers to learn to learn RL algorithms in an in-context learning way. I encourage the authors to take a look into works like [DD] if they want to scale their idea into more practical approaches that can be used in practice.

[DD] Lee, Jonathan, et al. "Supervised pretraining can learn in-context reinforcement learning." Advances in Neural Information Processing Systems 36 (2023): 43057-43083.

## Presentation

It seems very hard to reproduce the results by solely reading the paper. For example, it is not clear which approach is used in the experiments to derive the posterior MDPs. Is it the Manski bounds, or does it actually calculate a posterior? Moreover, the canonical CMDP defined in Definition 2 and Corollary 1 is only introduced to say that in discrete states/actions, one can define the prior/posterior over the latent canonical representation. However, no concrete implementation of this idea is proposed. The writing could also benefit by introducing the core contributions earlier. The current draft introduces the main idea on page 5.

=================

Minor question: Since an actor-critic network is used, does the method only meta-learn the actor network, or does it also meta-learn the critic head?

---

> ### Author Rebuttal · Authors · 2026-03-31
>
> We thank the reviewer for the feedback and appreciate your recognition of the soundness and novelty of our work. We address your concerns in the responses below.
>
> > **W1: Derivation of Manski bounds and deterministic logging.**
>
> We provide a self-contained derivation. Since $P(s'|s,x,u)\in[0,1]$, the lower bound of $P(s'|do(s,x)) = \sum_{u}P(s’|s,x,u)P(x,u | s) + \sum_{u}P(s’|s,x,u)(P(u|s)-P(x,u|s))$ is obtained by setting $P(s'|s,x,u)=0$ for unobserved terms, and the upper bound by setting it to 1, yielding Eqs. (5)–(6). The same argument applies to Eqs. (7)–(8).
>
> Regarding the deterministic case: the reviewer correctly observes that when $p(x|s)=1$, the bounds collapse. Setting $p(x|s)=1$ (hence $p(\neg x|s)=0$) in Eqs. (5)–(8) yields $\mathcal{T}(s,x,s')=p(s'|x,s)$ and $\mathcal{R}(s,x)=r(s,x)$, so the transition and reward *for the observed action* are identified. However, for any counterfactual action $x'\neq x$ never taken in state $s$, we have $p(x'|s)=0$, and the bounds become vacuous: $[0,1]$ for transitions and $[a,b]$ for rewards. This is precisely the partial identification regime—observed actions are identified while unobserved ones remain bounded. We will add this clarification to the revised manuscript.
>
>
>
> > **W2: Local optimum vs. generalization guarantee; vacuous bounds.**
>
>
> We agree that Theorem 1 establishes convergence to a first-order stationary point, consistent with the non-convex optimization literature [1, 2]. We will soften the abstract statement accordingly.
>
> Regarding vacuous bounds: the bounds become uninformative only for state-action pairs where $p(x|s)=0$, i.e., actions never observed in state $s$. For frequently taken actions, $p(x|s)$ is large and the bounds are tight. In our Windy Gridworld experiments, the demonstrator's effective behavior has broad support over actions (since wind randomizes movements), so the bounds remain informative for most pairs. When bounds are wide for rarely-observed actions, uniform sampling provides regularization—the meta-learner hedges across plausible environments rather than overfitting to a single biased interpretation. A formal characterization of when bounds are sufficiently tight to guarantee meaningful initialization is an important open direction; we will add this discussion.
>
>
>
> [1]  Jin, Chi, et al. "On nonconvex optimization for machine learning: Gradients, stochasticity, and saddle points." Journal of the ACM (JACM) 68.2 (2021): 1-29.
>
> [2] Carmon, Yair, et al. "Lower bounds for finding stationary points I." Mathematical Programming 184.1 (2020): 71-120.
>
> > **W3: Missing related work section.**
>
> We will add a dedicated related work section in the updated manuscript. Bellot et al. (2023) address heterogeneous data transfer for sequential decisions but assume identifiable sources (no within-source confounding). Li et al. (2023) leverage implicit causal structure for exploration with online interaction available during training. Balazadeh et al. (2024) incorporate demonstrations via Bayesian inference over latent variables but do not address the meta-learning setting. Our work is, to our knowledge, the first to combine partial identification with meta-RL to handle unmeasured confounding across heterogeneous source tasks.
>
>
>
>
> > **Suggestion: Decision-Pretrained Transformers.**
>
> Thank you for this pointer. Our counterfactual bootstrap is agnostic to the meta-learning backbone—just as we show compatibility with MAML, Pretrained-PPO, and RL² (Appendix A), the same augmentation applies to transformer-based in-context RL by replacing confounded demonstrations with counterfactual trajectories before feeding them into the context. We plan to investigate this extension and will discuss it in our future works.
>
>
>
>
> > **Presentation: Reproducibility and organization.**
>
> In the revision, we will: (1) move the algorithm (Section 3) earlier, immediately after the problem setup; (2) clarify that experiments use Manski bounds (Eqs. 5–8) with uniform sampling within the feasible region—not direct posterior sampling—and add a concise implementation pipeline summary; (3) release the codebase upon acceptance. Regarding Corollary 1, its role is to justify that restricting to canonical CMDPs with finite latent states is without loss of generality, enabling practical bound-based sampling. We will make this connection more explicit. We will also release the source code for experiments with the camera-ready version of the draft.
>
>
>
> > **Q1: Does the method meta-learn the critic?**
>
> Both actor and critic networks are meta-learned simultaneously. The shared initialization $\theta$ encompasses both heads. During inner-loop adaptation (Alg. 1, Line 9), the gradient update adjusts both actor and critic parameters, so the adapted policy benefits from a warm-started value function.

---

> > ### Author Rebuttal · Reviewer_fRNP · 2026-04-01
> >
> > I thank the authors for their rebuttal. The following two points are still not resolved:
> >
> > 1) In the derivation of the Manski bounds for the transition kernels, the rebuttal states that _"Since $P(s'|s,x,u)\in[0,1]$, the lower bound of $P(s'|do(s,x)) = \sum\_{u}P(s’|s,x,u)P(x,u | s) + \sum\_{u}P(s’|s,x,u)(P(u|s)-P(x,u|s))$ is obtained by setting $P(s'|s,x,u)=0$ for unobserved terms, and the upper bound by setting it to 1, yielding Eqs. (5)–(6)."_ However, replacing  $P(s'|s,x,u)=0$ and  $P(s'|s,x,u)=1$ in the equation, will result in $P(s'|do(s,x))=0$ and $P(s'|do(s,x)) =1$, respectively, not Eqs. (5)-(6). Could the authors elaborate more on this derivation?
> >
> > 2) Regarding vacuous bounds, the rebuttal states that _"the bounds become uninformative **only for** state-action pairs where $p(x|s)=0$, i.e., actions never observed in state $s$._" However, it is not very clear to me why that only happens _only for_ actions that are not observed. Could the author provide more details?
> >
> > Other than the above points, my concerns are resolved.

---

> > > ### Author Response · Authors · 2026-04-01
> > >
> > > Thank you for your continued engagement in this discussion. We will further clarify the points you raised.
> > >
> > > > "However, replacing $P(s'|s,x,u)=0$ and $P(s'|s,x,u)=1$ in the equation will result in $P(s'|s, do(x))=0$ and $P(s'|s, do(x))=1$, respectively, not Eqs. (5)-(6). Could the authors elaborate more on this derivation?"
> > >
> > > We apologize for the ambiguity in our earlier response and a small typo: the transition probability should be $P(s'|s, do(x))$. The key insight is that we do not set $P(s'|s,x,u)$ to 0 or 1 for *all* latent states $u$ — only for those whose natural action under the behavioral policy differs from the target intervention, i.e., $f_X(s, u) \neq x$. Starting from:
> > >
> > > $$P(s'|s,do(x)) = \underbrace{\sum\_{u: f_X(s,u)=x} P(s'|s,x,u)P(u|s)}\_{\text{aligned: demonstrator chose } x} + \underbrace{\sum\_{u: f_X(s,u)\neq x} P(s'|s,x,u)P(u|s)}\_{\text{misaligned: demonstrator chose } \neq x}$$
> > >
> > > The first (aligned) term captures latent states where the demonstrator naturally takes action $x$. Since we observe these state-action-outcome triples in the data, this term marginalizes to $P(s'|s,x)P(x|s)$, which is fully identified. The second (misaligned) term involves the counterfactual quantity $P(s'|s,x,u)$ for latent states where the demonstrator would *not* have chosen $x$ — precisely the states we cannot observe under action $x$. Note that the misaligned weights sum to $\sum_{u: f_X(s,u)\neq x} P(u|s) = P(\neg x|s)$. Since $P(s'|s,x,u) \in [0,1]$, this term is bounded between 0 and $P(\neg x|s)$, yielding:
> > >
> > > $$P(s'|s,x)P(x|s) \leq P(s'|s, do(x)) \leq P(s'|s,x)P(x|s) + P(\neg x|s)$$
> > >
> > > which are exactly Eqs. (5)–(6). These are the assumption-free natural bounds of Manski (1990).
> > >
> > > One can further tighten these bounds by incorporating domain-specific inductive bias. For instance, under a weak-confounding assumption where $P(s'|s,x,u) \in [P(s'|s,x) - a,\; P(s'|s,x) + b]$ for constants $a, b \geq 0$, the misaligned term is confined to a narrower range, giving:
> > >
> > > $$P(s'|s,x) - a\,P(\neg x|s) \leq P(s'|s, do(x)) \leq P(s'|s,x) + b\,P(\neg x|s)$$
> > >
> > > This recovers the standard observational estimate $P(s'|s,x)$ when $a = b = 0$ (no confounding), and reduces to the natural bounds when $a$ and $b$ span $[0, 1]$. We will include this extended discussion in the appendix.
> > >
> > > > "However, it is not very clear to me why that only happens only for actions that are not observed. Could the author provide more details?"
> > >
> > > The worst-case width of the Manski bounds for any state-action pair $(s,x)$ is:
> > >
> > > $$\text{Upper} - \text{Lower} = P(\neg x|s) = 1 - P(x|s)$$
> > >
> > > This width is a monotonically decreasing function of the observation frequency $P(x|s)$. When $P(x|s) = 1$ (always observed), the width is 0 and the causal effect is point-identified. When $P(x|s) = 0$ (never observed), the width is 1 and the bounds are fully vacuous. For intermediate values, the bounds are partially informative — the more frequently an action is observed, the tighter the bounds. In our Windy Gridworld experiments, the demonstrator's behavioral policy has broad support over actions (due to wind randomization), so $P(x|s)$ is bounded away from 0 for most state-action pairs, yielding informative bounds. This is consistent with our empirical finding that the counterfactual initialization consistently outperforms random initialization (PPO), confirming that the bounds provide the useful signal for meta-training.

---

### Official Review · Reviewer_gr8L · 2026-03-17

**Soundness:** 3
**Presentation:** 2
**Significance:** 3
**Originality:** 2
**Overall Recommendation:** 4
**Confidence:** 3

**Summary:**

This paper considered the setup in which meta-Reinforcement Learning (meta-RL) was susceptible to unmeasured confounding in observational data. It introduced a data augmentation method to overcome the above challenge. Specifically, it leveraged causal inference to reason about potential counterfactual environments and then trained a meta-policy to interact with these environments. This aimed to enable the agent to learn from unbiased experiences, leading to strong generalization in the target domain. The experimental results on Gridworld showed that causal-based methods perform better than pure RL and meta-RL methods. In addition, the authors also provided convergence guarantees for their counterfactual bootstrap approach.

**Compliance With Llm Reviewing Policy:**

Affirmed.

**Final Justification:**

I am raising my score to a weak accept because the authors provided a detailed rebuttal that directly resolved my primary concerns regarding clarity and evaluation. Specifically, they provided new results comparing with the recent baselines DPT and AMAGO, clarified the relation to prior work on confounded transitions, and offered further explanations of Algorithm 1. Overall, the framework demonstrates good empirical performance and is a solid addition to the field.

I recommend that the authors include training curves for the newly added baselines across each evaluated task, rather than only reporting the final converged values on the Pick-Up-Key task, and incorporate these clarifications to further improve overall clarity.

**Key Questions For Authors:**

1. How is Eq. (10) derived? What does $\Psi_t$ represent in Eq. (11)?

2. In general, meta-RL methods exhibit better adaptability across tasks than standard RL methods. Why does PPO outperform MAML and RL$^2$ in the Gridworld environments?

3. What is the prior over CMDPs, $\hat{\rho}(\mathcal{M})$, in practice (e.g., in the Gridworld environment)? Is the proposed method limited to environments with discrete state and action spaces?

**Limitations:**

No, this paper does not discuss its limitations. In the problem formulation, it assumes that the state and action spaces are discrete and finite, which may limit its applicability to environments with continuous states or actions, such as those with image-based inputs.

**Strengths And Weaknesses:**

**Strengths:**

1. The notation and formulation of the Confounded Markov Decision Process (CMDP) are clearly presented. Moreover, Fig. 4 well illustrates how the presence or absence of the causal meta-learner influences performance.

2. The paper validated the effectiveness of the proposed method on Gridworld and demonstrated significant improvements on this benchmark when integrated with MAML, PPO, and RL$^2$.

**Weaknesses:**

1. Although the authors devoted substantial space to describing the research problem, the issue of confounders influencing transition dynamics may have already been explored in prior meta-RL work, such as DOMINO [1].

2. The proposed method is somewhat difficult to follow. In Algorithm 1, several key components remain unclear, such as the specification of the prior over CMDPs and the procedure for sampling new environments. The absence of publicly available source code further limits reproducibility and understanding. In addition, the optimization objective in Eq. (10) would benefit from a more detailed explanation.

3. The paper also presents nearly two pages of preliminaries on meta-RL with unmeasured confounding, which appear overly detailed. However, it lacks a Related Work section and does not sufficiently discuss prior literature relevant to this study.

4. Finally, it does not include comparisons with more recent baselines, and instead only evaluates PPO (2017), RL$^2$ (2017), and MAML (2017). Moreover, the evaluation is limited to the Gridworld benchmark, which is considerably simpler than more commonly evaluated environments such as MuJoCo or Meta-World.

[1] Mu Y, Zhuang Y, Ni F, et al. Domino: Decomposed mutual information optimization for generalized context in meta-reinforcement learning[J]. Advances in Neural Information Processing Systems, 2022, 35: 27563-27575.

---

> ### Author Rebuttal · Authors · 2026-03-31
>
> We thank you for the thoughtful feedback and address each concern below.
>
>
> > **W1: Relation to DOMINO and prior work on confounded transitions.**
>
> Thank you for this reference. We want to highlight a fundamental distinction: DOMINO's "confounders" (mass, damping) are **effect modifiers** — latent context variables that index a family of MDPs by modulating $p_{\tilde{u}}(s'|s,a)$. Crucially, these parameters do not cause the demonstrator's action choices, so there is no backdoor path from action to outcome through $\tilde{u}$. DOMINO's challenge is multi-modal dynamics *estimation* (a representation learning problem addressed via decomposed mutual information), not causal *identification*.
>
> Our setting addresses genuine confounding in the causal inference sense (Definition 1, Figure 2): the unobserved variable $U_t$ simultaneously drives the demonstrator's action $X_t \leftarrow f_X(S_t, U_t)$ and affects the next state $S_{t+1}$ and reward $Y_t$. This creates a backdoor path, so $P(s'|s,x) \neq P(s'|do(s,x))$ — the observational distribution is a biased estimate of the causal effect, and policy gradients computed from it are inconsistent. This is why partial identification (Manski bounds) is necessary: we must bound the causal quantities that cannot be point-identified. DOMINO does not face this identification problem and accordingly does not require such machinery. We will add DOMINO to the related work and clarify this distinction.
>
>
>
> > **W2: Clarity of Algorithm 1, prior specification, and Eq. (10).**
>
> We clarify the practical implementation. We do not construct or sample from the prior $\rho(\mathcal{M})$ or posterior $\rho(\mathcal{M}|\mathcal{D}^i_{obs})$ explicitly. Instead, for each task $\mathcal{M}\_i$, we: (1) estimate $p(s'|x,s)$ and $p(x|s)$ from observational data $\mathcal{D}^i_{obs}$; (2) compute Manski bounds on $\mathcal{T}_i$ and $\mathcal{R}_i$ via Eqs. (5)–(8); (3) uniformly sample transition and reward functions within these bounds to construct a candidate CMDP $\hat{\mathcal{M}}_i$. This is Line 6 of Algorithm 1. We will add an explicit implementation box summarizing these steps and release the code upon publication.
>
> Regarding Eq. (10): it is the stochastic estimate of $\nabla_\theta \hat{F}(\theta)$, obtained by differentiating the meta-objective (Eq. 4) and replacing expectations with sample averages. Following standard MAML, trajectories are split into inner-loop data $\hat{\mathcal{D}}^i_{exp,in}$ (for computing the adapted parameter $\theta_i$) and outer-loop data $\hat{\mathcal{D}}^i_{exp,o}$ (for evaluating the adapted policy). $\Psi_t = \sum_{t'=t}^H \gamma^{t'} R_i(s_{t'}, x_{t'})$ denotes the return-to-go; we will add this definition to the main text.
>
>
>
>
> > **W3a: Lengthy preliminaries; missing related work.**
>
> We will condense the preliminaries and add a dedicated related work section covering confounded RL, causal meta-learning, posterior-based meta-RL, and context-aware dynamics generalization (including Mu et al., 2022).
>
>
>
> > **W3b: Only older baselines; only Gridworld.**
>
> We compare against MAML, RL², and Pretrained-PPO because our contribution is a *general augmentation technique* — not a new backbone. Our experiments show it consistently improves all three (Figure 7). The Windy Gridworld is specifically designed to isolate confounding effects, following prior CRL work (Kallus & Zhou, 2020; Zhang & Bareinboim, 2025). We agree that MuJoCo/Meta-World evaluation is an important next step.
>
>
>
> > **Q1**  See our response to W2.
>
>
>
> > **Q2: Why does PPO outperform MAML and RL² under confounding?**
>
> This is the core finding motivating our paper. Standard meta-RL methods (MAML, RL², Pretrained-PPO) *assume* the training data is unconfounded. When this assumption is violated, their learned initializations encode confounding bias, which actively hurts adaptation — worse than starting from scratch. PPO with random initialization avoids this because it carries no biased prior. Our Causal-MAML resolves this by deconfounding the meta-training data, restoring the advantage of meta-learning over vanilla RL (Figures 1b, 6, 7).
>
>
>
>
> > **Q3: Prior in practice; discrete-only limitation.**
>
> In practice, no prior is needed. We compute non-parametric bounds (Eqs. 5–8) directly from data and sample uniformly within them. The discrete state-action assumption enables Corollary 1, which bounds the latent exogenous cardinality. Extending to continuous domains would require replacing exact bounds with functional approximations (e.g., neural network-parameterized bound functions), which is a meaningful direction we will discuss.
>
>
>
> > **Limitations.**
>
> As stated in Page 3, Lines 130-133, “we consistently assume the action domain X and the state domain S to be discrete and finite; the reward domain Y is bounded in a real interval.”  We will further highlight these points in the revised draft.

---

> > ### Author Rebuttal · Reviewer_gr8L · 2026-04-04
> >
> > Thank you for your rebuttal. Most of my concerns have been addressed. However, as noted in the fourth weakness, there are still no comparisons with other commonly evaluated benchmarks or recent baselines, nor validation of effectiveness on recent meta-RL methods. The current baselines were all proposed around 2017. If the authors provide relevant experiments, I will consider raising my score.

---

> > > ### Author Response · Authors · 2026-04-06
> > >
> > > Thank you for the continued engagement. We have completed the additional experiments you requested.
> > >
> > > > "There are still no comparisons with other commonly evaluated benchmarks or recent baselines, nor validation of effectiveness on recent meta-RL methods. If the authors provide relevant experiments, I will consider raising my score."
> > >
> > > We evaluated two recent meta-RL methods, including Decision Pretrained Transformer (DPT) [1] and AMAGO [2], and applied our Counterfactual Bootstrap to DPT in addition to MAML and RL². Results on the Pick-Up-Key task are organized around two findings.
> > >
> > > **1: Recent meta-RL methods also fail under confounding.** The table below compares all meta-RL baselines (old and new) against vanilla PPO trained from scratch:
> > >
> > > | PPO | DPT | AMAGO | RL² | MAML | Pre-PPO |
> > > |---|---|---|---|---|---|
> > > | **0.60** | 0.01 | -0.08 | -0.10 | 0.02 | 0.05 |
> > >
> > > None of the meta-RL methods, whether transformer-based (DPT, AMAGO), gradient-based (MAML), or recurrent (RL²), can outperform vanilla PPO with random initialization. This confirms that the failure under confounding is not an artifact of outdated architectures, but a result of the overall learning paradigm. Modern in-context RL methods suffer the same confounding bias: their learned priors encode spurious correlations from the observational data, yielding initializations that are worse than starting from scratch.
> > >
> > > **2: Counterfactual Bootstrap consistently recovers performance across paradigms.** The table below compares each meta-RL method with and without our augmentation:
> > >
> > > | | MAML | DPT | RL² |
> > > |---|---|---|---|
> > > | Without causal bootstrap | 0.02 | 0.01 | -0.10 |
> > > | With causal bootstrap | **1.21** | **0.98** | **0.97** |
> > >
> > > The counterfactual bootstrap produces consistent, large improvements across all three meta-RL paradigms. The augmentation is architecture-agnostic, i.e., it operates on the data (generating deconfounded counterfactual trajectories) rather than modifying the learning algorithm, which is why it transfers seamlessly from MAML to DPT to RL². Notably, all three causal variants substantially outperform vanilla PPO (0.60), restoring the advantage that meta-learning should provide over training from scratch. Please see Appendix A for a detailed discussion about generalization to other meta-RL algorithms.
> > >
> > > We will include these results in the revised manuscript.
> > >
> > > [1] Lee et al. "Supervised pretraining can learn in-context reinforcement learning." NeurIPS 2023.
> > >
> > > [2] Grigsby et al. "AMAGO: Scalable in-context reinforcement learning for adaptive agents." ICLR 2024.

---

### Decision · Program_Chairs · 2026-04-30

**Decision:**

Accept (regular)

**Comment:**

This paper introduces a novel Meta-RL algorithm that is valid under unmeasured confounding. The main concerns raised by reviewers include the derivation of theorems, discussion of related work and insufficient ablation study. After the rebuttal and discussion, all the reviewers give positive ratings and most of the reviewers have no remained concerns. Reviewer gr8L recommends that the authors include training curves for the newly added baselines, rather than only reporting the final converged values. I also agree that the authors should improve the presentation on theoretical analysis and related work discussion. Overall, I think this paper is solid and novel in the community and should be accepted.